# First SFT, Second RL, Third UPT: Continual Improving Multi-Modal LLM Reasoning via Unsupervised Post-Training

**Lai Wei**[1,2,*]    **Yuting Li**[1]    **Chen Wang**[2]    **Yue Wang**[2]    **Linghe Kong**[1]
**Weiran Huang**[1,3,†]    **Lichao Sun**[4]

[1] School of Computer Science, Shanghai Jiao Tong University
[2] Zhongguancun Academy
[3] Shanghai Innovation Institute    [4] Lehigh University

## Abstract

Improving Multi-modal Large Language Models (MLLMs) in the post-training stage typically relies on supervised fine-tuning (SFT) or reinforcement learning (RL), which require expensive and manually annotated multi-modal data–an ultimately unsustainable resource. This limitation has motivated a growing interest in unsupervised paradigms as a third stage of post-training after SFT and RL. While recent efforts have explored this direction, their methods are complex and difficult to iterate. To address this, we propose MM-UPT, a simple yet effective framework for unsupervised post-training of MLLMs, enabling continual self-improvement without any external supervision. The training method of MM-UPT builds upon GRPO, replacing traditional reward signals with a self-rewarding mechanism based on majority voting over multiple sampled responses. Our experiments demonstrate that such training method effectively improves the reasoning ability of Qwen2.5-VL-7B (e.g., 66.3%→72.9% on MathVista, 62.9%→68.7% on We-Math), using standard dataset without ground truth labels. To further explore scalability, we extend our framework to a data self-generation setting, designing two strategies that prompt the MLLM to synthesize new training samples on its own. Additional experiments show that combining these synthetic data with the unsupervised training method can also boost performance, highlighting a promising approach for scalable self-improvement. Overall, MM-UPT offers a new paradigm for autonomous enhancement of MLLMs, serving as a critical third step after initial SFT and RL in the absence of external supervision. Our code is available at https://github.com/waltonfuture/MM-UPT.

## 1    Introduction

Multi-modal Large Language Models (MLLMs) have achieved remarkable performance on a variety of vision-language tasks, ranging from image captioning to visual reasoning [22, 51, 67, 73, 81, 83]. The dominant paradigm for improving MLLMs in the post-training stage typically involves supervised fine-tuning (SFT) and reinforcement learning (RL) [2, 42, 47, 59, 65, 77]. However, both SFT and RL rely on large volumes of high-quality and annotated multi-modal data, such as image captions, visual reasoning traces, verifiable ground truth answers, and human preference signals. As real-world tasks

---

[*]Email: waltonfuture@sjtu.edu.cn
[†]Correspondence to Weiran Huang (weiran.huang@outlook.com).

39th Conference on Neural Information Processing Systems (NeurIPS 2025).

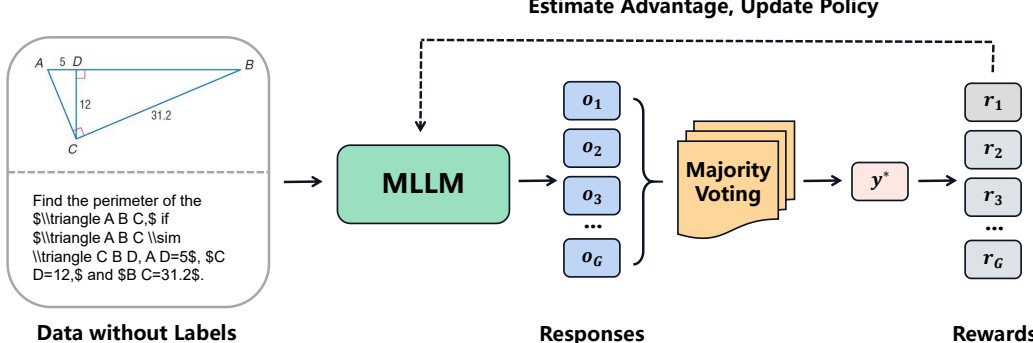

Figure 1: Overview of the MM-UPT framework. Given an unlabeled multi-modal input, the MLLM samples multiple responses, and uses majority voting to determine the pseudo-label. The MLLM is then updated via GRPO, enabling self-improvement without external supervision.

grow in complexity and quantity, a critical challenge emerges: curating and annotating high-quality data at scale becomes increasingly impractical. To overcome this data-dependency, a paradigm shift is required towards a third stage of post-training beyond SFT and RL, dedicated to the continual self-improvement of MLLMs through synthetic and unlabeled data. We formalize this third-stage paradigm as *Unsupervised Post-Training*.

Previous works have studied the use of MLLMs themselves to generate synthetic instruction data for self-improvement through offline training techniques like SFT and DPO [6, 23, 39–41, 64, 74]. These approaches typically involve complex pipelines with multiple stages, such as data generation, verification, and filtering, which are hard to iterate online. Fortunately, recent studies demonstrate notable success using online reinforcement learning (e.g. GRPO [37]) with verifiable rewards to enhance the reasoning capabilities of MLLMs [7, 31, 53]. A concurrent work, TTRL [84], further extends this line by applying GRPO on test-time scaling of LLMs. It is promising that online RL enables models to continuously improve, thus acquiring novel reasoning abilities that exceed corresponding base models' capacity.

Motivated by these insights, we propose MM-UPT (Multi-Modal Unsupervised Post-Training), an easy-to-implement framework for unsupervised post-training in MLLMs. As illustrated in Figure 1, our approach adapts the online reinforcement learning method GRPO [37], which is known for its stability and scalability. The core challenge in applying GRPO to an unsupervised setting is the absence of ground-truth labels for reward calculation. To overcome this, MM-UPT works by deriving implicit reward signals via majority voting over multiple sampled responses. In particular, majority voting aggregates multiple responses and selects the most frequent one, which has been widely used and shown effective to improve model performance [44, 49, 84]. Thus, we adopt the majority-voted answer to serves as a dynamic pseudo-label in GRPO: responses that align with this consensus receive a positive reward, while divergent ones are penalized. This process effectively encourages the model to bootstrap its own high-confidence knowledge, promoting the generation of stable and consistent answers without relying on any external supervision or reward models.

Beyond learning from existing unlabeled data, MM-UPT further extends to a data self-generation setting that enhances scalability under the unsupervised paradigm. Specifically, we design two strategies that allow the MLLM itself to synthesize new training samples: (1) *in-context synthesizing*, where the model generates new questions conditioned on original examples (image, question, and answer); and (2) *direct synthesizing*, where the model generates questions solely from the given image. These self-generated samples are then used within the same unsupervised reinforcement learning method, enabling continual self-improvement even in the absence of human-created questions.

In our experiments, we focus on the domain of multi-modal reasoning, which is widely focused and inherently challenging. We explore two key scenarios for constructing unlabeled data assuming that labels are not available: (1) using human-created questions without ground-truth labels, and (2) employing synthetic questions generated by the model itself, inherently lacking ground-truth labels. This setup allows us to examine both the effectiveness of unsupervised training on existing data and self-generated data in enhancing reasoning performance. We evaluate MM-UPT across a range of reasoning benchmarks and observe notable performance improvements over the base models

(e.g., 66.3%→72.9% on MathVista, 62.9%→68.7% on We-Math using Qwen2.5-VL-7B [1]) in the first scenario. Our method also outperforms previous baseline methods, and is even competitive with supervised GRPO, underscoring the effectiveness of MM-UPT as a self-improving training strategy. As for the second scenario, we find that models trained on unlabeled synthetic data achieve performance competitive with those trained on the original unlabeled dataset, revealing a viable path for scalable self-improvement. Additionally, our deeper analysis reveals a clear trade-off in MM-UPT: it improves accuracy by reinforcing high-confidence knowledge but reduces response diversity and requires sufficient initial competence to prevent error amplification.

Our main contributions are summarized as follows:

- We are the first to formalize a three-stage post-training paradigm for MLLMs, with Unsupervised Post-Training (UPT) as the critical third stage enabling continual improvement without external supervision. We instantiate this stage with MM-UPT, a simple and effective framework using majority voting as a self-rewarding signal in online reinforcement learning.
- Extensive experiments on multi-modal reasoning tasks demonstrate the effectiveness of majority voting as a pseudo-reward estimation for unsupervised training.
- We extend MM-UPT to utilize synthetic data generated by the MLLM itself, and find that training the MLLM on such data leads to notable performance gains. This reveals a promising path toward efficient and scalable self-improvement in unsupervised post-training.

## 2 Related Works

**Self Improvement.** High-quality data obtained from human annotations has been shown to significantly boost the performance of LLMs across a wide range of tasks [12, 18, 32]. However, such high-quality annotated data may be exhausted in the future. This presents a substantial obstacle to the continual learning of advanced models. As a result, recent research has shifted toward self-improvement, leveraging data generated by the LLM itself without any external supervision [8, 16, 29, 54, 84]. Several following works also explore self-improvement in the multi-modal domain [6, 11, 39, 64, 72, 74]. Genixer [74] firstly introduces a self-improvement pipeline including complex data generation and filtering for SFT. STIC [6] and SENA [40] construct preference data pairs for DPO [36] in a self-supervised manner, focusing on enhancing perceptual capabilities. In contrast to these approaches which are complex and hard to scale, the key distinction is that our work leverages online reinforcement learning using GRPO [12] with simple yet effective data synthesizing strategies at the post-training stage, which can be more scalable for self-improvement without reliance on any external supervision. In addition, none of these previous methods focus on multi-modal reasoning tasks, which are considered challenging for current models.

**Multi-modal Reasoning.** Recently, the reasoning abilities of MLLMs have become a central focus of research [27, 53, 56, 82]. In contrast to traditional LLM-based reasoning [12, 28, 62] that primarily relies on text, multi-modal approaches must both process and interpret visual inputs, significantly increasing the complexity of tasks such as geometric problem-solving and chart interpretation [3, 30, 75]. Several works in this field have sought to collect or synthesize a large scale of multi-modal reasoning data [5, 33, 38, 70]. Notably, the recent emergence of o1-like reasoning models [19] represents an initial step toward activating the slow-thinking capabilities of MLLMs, as demonstrated by several SFT-based methods, such as LLaVA-CoT [55], MAmmoTH-VL [13], and Mulberry [60]. Moreover, some concurrent works have further explored reinforcement learning approaches, particularly GRPO [37], in the post-training stage of MLLMs to enhance performance on multi-modal reasoning tasks [7, 31, 34, 48, 78]. While these supervised post-training methods have demonstrated promising results, our work explores a different direction by focusing on totally unsupervised post-training of MLLMs to self-improve the reasoning abilities.

## 3 The Framework of Multi-Modal Unsupervised Post-Training

Existing post-training techniques for MLLMs, such as supervised fine-tuning (SFT) and reinforcement learning (RLHF or RLVR), rely heavily on labeled data or external reward models. While these approaches have proven effective, their reliance on external supervision makes continual improvement unsustainable due to the high cost and limited scalability of manual annotation. To overcome this limitation, we formalize a new third-stage post-training paradigm that enables the model to self-

improve without access to any external supervision, such as ground-truth labels or additional reward models. We instantiate this paradigm with **MM-UPT** (Multi-Modal Unsupervised Post-Training), a simple yet effective framework designed to operate purely on unlabeled multi-modal data. The overview of our complete framework is shown in Figure 1.

## 3.1   Problem Formulation

Firstly, we formulate the problem of unsupervised post-training for MLLMs as follows: Given a well-trained multi-modal LLM $\pi_\theta$ and a collection of unlabeled multi-modal data $Q = \{(I_i, q_i)\}_{i=1}^N$, where $I_i$ represents an image and $q_i$ denotes a corresponding question, our goal is to improve the model's performance without access to any ground-truth answers or external supervision signals. This setting differs significantly from conventional supervised fine-tuning (SFT), reinforcement learning with verifiable rewards (RLVR), or reinforcement learning with human feedback (RLHF), which typically rely on labeled data $(I_i, q_i, y_i)$ or human preference data $(I_i, q_i, y_i^+, y_i^-)$, where $y_i$ denotes the answer of $q_i$ and $(y_i^+, y_i^-)$ denotes the preference pair of $q_i$. In contrast, we only allow to operate in a fully unsupervised manner for this setting, leveraging only the model's own responses to generate training signals. This presents significant challenges, as the model must learn to assess and improve its own outputs without any external guidance.

## 3.2   Training Method

To achieve the unsupervised training, MM-UPT introduces a self-rewarding mechanism using majority voting as pseudo-labels [49] based on the online reinforcement learning. In particular, MM-UPT is built upon the GRPO algorithm [37], which is widely used in the post-training stage of multi-modal LLMs. GRPO optimizes computational efficiency by eliminating the need for a separate value model; instead, it directly utilizes group-normalized rewards to estimate advantages. Specifically, for a question $q$ and the correlated image $I$ from the training dataset $Q$, GRPO samples a group of responses $O = \{o_i\}_{i=1}^G$ from the old policy $\pi_{old}$ and then optimizes the policy model by maximizing the following objective:

$$\mathcal{J}(\theta) = \mathbb{E}_{(q,I)\sim Q, \{o_i\}_{i=1}^G \sim \pi_{\theta_{old}}(O|q,I)}$$

$$\frac{1}{G}\sum_{i=1}^G \frac{1}{|o_i|}\sum_{t=1}^{|o_i|}\left\{\min\left[\gamma_{i,t}(\theta)\hat{A}_{i,t}, \text{clip}\left(\gamma_{i,t}(\theta), 1-\epsilon, 1+\epsilon\right)\hat{A}_{i,t}\right] - \beta\mathbf{D}_{KL}\left[\pi_\theta\|\pi_{ref}\right]\right\},$$

where $\gamma_{i,t}(\theta) = \frac{\pi_\theta(o_{i,t}|q,o_{i,<t})}{\pi_{\theta_{old}}(o_{i,t}|q,o_{i,<t})}$, $\pi_{ref}$ represents the reference model, and the term $D_{KL}$ introduces a KL divergence constraint to limit how much the model can deviate from this reference. The advantage estimate $\hat{A}_i$ measures how much better the response $o_i$ is compared to the average response, which is computed using a group of rewards $\{r_1, r_2, \ldots, r_G\}$ for the responses in set $O$: $\hat{A}_i = \frac{r_i - \text{mean}(\{r_1, r_2, \ldots, r_G\})}{\text{std}(\{r_1, r_2, \ldots, r_G\})}$.

In the above standard GRPO formulation [12], the reward is computed in a supervised manner based on labels for each response in $O = \{o_i\}_{i=1}^G$. Shifting towards our unsupervised setting, where no ground-truth labels are available, one feasible way is to construct pseudo-labels to calculate the reward for GRPO. Motivated by [16, 49, 84], we use majority voting over the group of sampled responses $O$ to serve as pseudo-labels. Majority voting selects the most frequent answer among the sampled responses $O$ and has proven to be a simple yet effective technique [49, 84], making it suitable for deriving good pseudo-reward signals. Specifically, we first extract answers from the responses $O = \{o_i\}_{i=1}^G$ using an rule-based answer extractor [15] $E(\cdot)$, resulting in $\hat{Y} = E(O) = \{\hat{y}_i\}_{i=1}^G$. Then, the majority-voted answer $y^*$ can be obtained by:

$$y^* = \arg\max_{y \in \hat{Y}} \sum_{i=1}^G \mathbb{I}[y = \hat{y}_i], \tag{1}$$

where $\mathbb{I}[\cdot]$ is the indicator function. The reward $r_i$ is then determined based on the $y^*$:

$$r_i = \begin{cases} 1, & \text{if } \hat{y}_i = y^*, \\ 0, & \text{otherwise.} \end{cases} \tag{2}$$

---

**Algorithm 1** The training method of MM-UPT

---

1: **Input:** Current policy $\pi_\theta$, old policy $\pi_{\theta_{old}}$, unlabeled training dataset $Q$, Group size $G$, reference model $\pi_{ref}$, clip parameter $\epsilon$, KL penalty coefficient $\beta$, answer extractor $E(\cdot)$.
2: **for** each sample $(I, q) \sim Q$ **do**
3:     Sample group of responses: $\{o_i\}_{i=1}^G \sim \pi_{\theta_{\text{old}}}(o \mid I, q)$;           // Sample multiple responses
4:     Extract answers: $\hat{Y} = E(O) = \{\hat{y}_i\}_{i=1}^G$;
5:     Determine majority vote: $y^* \leftarrow \arg\max_{y \in \hat{Y}} \sum_{i=1}^G \mathbb{I}[y = \hat{y}_i]$;     // Select the most frequent answer
6:     Compute pseudo-rewards: $r_i \leftarrow \mathbb{I}[\hat{y}_i = y^*]$;         // Reward based on majority agreement
7:     Compute advantage estimates: $\hat{A}_i \leftarrow \frac{r_i - \text{mean}(\{r_1, r_2, \dots, r_G\})}{\text{std}(\{r_1, r_2, \dots, r_G\})}$;
8:     Compute GRPO objective:
9:     $\mathcal{J}(\theta) \leftarrow \frac{1}{G} \sum_{i=1}^G \frac{1}{|o_i|} \sum_{t=1}^{|o_i|} \left\{ \min \left[ \gamma_{i,t}(\theta) \hat{A}_i, \text{clip}\left(\gamma_{i,t}(\theta), 1-\epsilon, 1+\epsilon\right) \hat{A}_i \right] - \beta \mathrm{D}_{KL}[\pi_\theta \| \pi_{ref}] \right\}$
10:     where $\gamma_{i,t}(\theta) = \frac{\pi_\theta(o_{i,t} | I, q, o_{i,<t})}{\pi_{\theta_{\text{old}}}(o_{i,t} | I, q, o_{i,<t})}$;
11:     Update policy parameters: $\theta \leftarrow \theta - \nabla_\theta \mathcal{J}_{GRPO}(\theta)$;
12:     Update old policy: $\theta_{\text{old}} \leftarrow \theta$;
13: **end for**
14: **return** $\pi_\theta$

---

In this way, we compute pseudo-rewards via majority voting and apply standard GRPO to update the MLLM. This majority-based reward encourages the model to converge toward consistent, high-consensus responses, thereby enabling the model to further exploit its existing self-knowledge leveraging unlabeled data. The pipeline of our training method in MM-UPT is shown in Algorithm 1.

### 3.3 Synthetic Data

To further explore the scalability of unsupervised post-training, we extend our framework to a data self-generation setting, where the model is asked to synthesize new training samples on its own. In particular, we design two simple yet effective data synthesizing strategies as follows.

**In-Context Synthesizing.** Inspired by Self-Instruct [50], we construct a data generation pipeline that leverages in-context examples to guide the synthesis process. Each original example consists of an image, a question, and its corresponding answer. To synthesize new samples, we provide the model with the full triplet and instruct it to generate a new question that is semantically distinct from the original but relevant to the same image. This strategy helps generate task-relevant and meaningful variations of the original question, as well as ensure the quality of synthetic questions. During unsupervised post-training, the model then attempts to answer each newly generated question, and pseudo-labels are derived from the majority vote among its sampled responses, consistent with the mechanism described in Section 3.

**Direct Synthesizing.** In addition to in-context generation, we also explore a *direct synthesizing* strategy that further increases diversity. Here, the model receives only the image and is prompted to freely create a new question without any reference to the original question. This open-ended formulation encourages the model to generate a wider range of diverse and novel questions based solely on the visual input, rather than being constrained by the original task. Similar to the in-context setup, we perform unsupervised post-training on these synthetic samples using majority voting to define pseudo-rewards.

This setup enables the MLLM to expand the training corpus without any human annotations, thus achieving a fully autonomous self-improvement loop.

## 4 Experiments

We conduct extensive experiments to evaluate the effectiveness of MM-UPT across various multimodal LLMs, datasets, and benchmarks. Our experiments are designed to explore **two key scenarios**: (1) using human-created questions without ground-truth labels (Section 4.2), and (2) employing synthetic questions generated by the model itself, inherently lacking ground-truth labels (Section 4.3). Before presenting the experimental results, we first outline the baseline methods, evaluation benchmarks, and implementation details in the experimental setup as follows.

Table 1: Main results of **Scenario 1** on four multi-modal mathematical reasoning benchmarks. We report accuracy (%) for each method on MathVision, MathVerse, MathVista, and We-Math. All methods are conducted on the Qwen2.5-VL-7B backbone. MM-UPT outperforms other baseline methods, and is even competitive with supervised methods.

| Model and Methods | Unsupervised? | Training Data | MathVision | MathVerse | MathVista | We-Math | Avg |
|---|---|---|---|---|---|---|---|
| Qwen2.5-VL-7B | - | - | 24.87 | 43.83 | 66.30 | 62.87 | 49.47 |
| **+ GRPO** [37] | ✗ | Geometry3K | 28.32 | 46.40 | 69.30 | 68.85 | 53.22 |
| **+ GRPO** [37] | ✗ | GeoQA | 26.15 | 46.28 | 67.50 | 66.65 | 51.65 |
| **+ GRPO** [37] | ✗ | MMR1 | 29.01 | 45.03 | 71.40 | 67.24 | 53.17 |
| **+ SFT** [43] | ✗ | Geometry3K | 25.92 | 43.73 | 67.90 | 64.94 | 50.63 |
| **+ SFT** [43] | ✗ | GeoQA | 25.72 | 44.70 | 67.40 | 65.10 | 50.73 |
| **+ SFT** [43] | ✗ | MMR1 | 26.45 | 43.53 | 63.30 | 64.20 | 49.37 |
| **+ SRLM** [54] | ✓ | Geometry3K | 26.94 | 44.54 | 66.90 | 66.32 | 51.18 |
| **+ SRLM** [54] | ✓ | GeoQA | 25.16 | 44.62 | 66.30 | 65.00 | 50.27 |
| **+ SRLM** [54] | ✓ | MMR1 | 25.33 | 45.08 | 67.00 | 64.66 | 50.52 |
| **+ LMSI** [16] | ✓ | Geometry3K | 25.10 | 43.96 | 65.50 | 64.43 | 49.75 |
| **+ LMSI** [16] | ✓ | GeoQA | 25.49 | 43.50 | 66.60 | 63.51 | 49.78 |
| **+ LMSI** [16] | ✓ | MMR1 | 24.83 | 43.76 | 64.90 | 66.38 | 49.97 |
| **+ Genixer** [74] | ✓ | Geometry3K | 26.02 | 43.15 | 65.50 | 62.18 | 49.22 |
| **+ Genixer** [74] | ✓ | GeoQA | 25.30 | 44.11 | 66.80 | 64.25 | 50.12 |
| **+ Genixer** [74] | ✓ | MMR1 | 23.68 | 43.30 | 65.50 | 64.66 | 49.29 |
| **+ STIC** [6] | ✓ | Geometry3K | 25.39 | 42.92 | 65.20 | 62.99 | 49.13 |
| **+ STIC** [6] | ✓ | GeoQA | 23.49 | 42.87 | 64.30 | 63.62 | 48.57 |
| **+ STIC** [6] | ✓ | MMR1 | 23.78 | 42.72 | 66.10 | 63.74 | 49.09 |
| **+ MM-UPT (Ours)** | ✓ | Geometry3K | 27.33 | 42.46 | 68.50 | 66.61 | **51.23** |
| **+ MM-UPT (Ours)** | ✓ | GeoQA | 27.07 | 43.68 | 68.90 | 68.22 | **51.97** |
| **+ MM-UPT (Ours)** | ✓ | MMR1 | 26.15 | 44.87 | 72.90 | 68.74 | **53.17** |

## 4.1 Experimental Setup

**Baseline Methods.** Several prior works have explored self-improvement in both LLMs and MLLMs. Note that we focus on unsupervised self-improvement, we do not compare with methods that rely on external models (e.g., GPT-4o [18]) for supervision [17, 64, 71, 79, 80]. Instead, we compare with several totally unsupervised methods: LMSI [16], SRLM [54], Genixer [74], and STIC [6]. In particular, LMSI corresponds to supervised fine-tuning with self-generated content selected by majority voting. SRLM uses the model itself as LLM-as-a-Judge [76] to provide its own rewards during DPO [36] training. Genixer prompts the MLLM to first self-generate an answer and then self-check it. STIC applies DPO where original images and good prompts are used to generate preferred answers, and corrupted images and bad prompts to produce rejected answers. Additionally, we also compare with GRPO [37] and rejection SFT [43], which are two strong supervised methods. The details of these baseline methods are shown in Appendix A.1.

**Benchmarks.** We evaluate our method on four popular multi-modal mathematical reasoning benchmarks: MathVision [45], MathVista [27], MathVerse [69], and We-Math [35]. These benchmarks offer comprehensive evaluations with diverse problem types, including geometry, charts, and tables, featuring multi-subject and meticulously categorized visual math challenges across various knowledge concepts and granularity levels. We provide more details in Appendix A.2.

**Implementation Details.** We adopt the EasyR1 [61] framework for multi-modal unsupervised post-training, which is based on GRPO. Specifically, we set the training episodes to 15, and use AdamW optimizer [24] with a learning rate of $1 \times 10^{-6}$, weight decay of $1 \times 10^{-2}$, and gradient clipping at a maximum norm of 1.0. The KL divergence constraint $\beta$ in GRPO is set to 0.01 to stabilize the training. The vision tower of the multi-modal model is also tuned without freezing. In our training, we use a rollout temperature of 0.7, which strikes a good balance: lower temperatures produce low-diversity outputs, while higher temperatures often lead to lower-quality outputs. Within each episode, we perform one rollout group ($G$=10) per data point. Other hyperparameters follow the default settings provided in the EasyR1 framework.

Table 2: Performance comparison of MM-UPT using different synthetic data generation strategies in **Scenario 2**. Both "In-Context Synthesizing" and "Direct Synthesizing" approaches yield significant improvements over the base model and perform competitively with the "Original Questions" on average, demonstrating the effectiveness of synthetic data for unsupervised self-improvement.

| Model and Methods | Dataset | MathVision | MathVerse | MathVista | We-Math | Avg |
|---|---|---|---|---|---|---|
| Qwen2.5-VL-7B | – | 24.87 | 43.83 | 66.30 | 62.87 | 49.47 |
| w/ Original Questions | Geo3K | 27.33 | 42.46 | 68.50 | 66.61 | 51.23 (**3.6%**↑) |
| w/ In-Context Synthesizing | Geo3K | 26.71 | 41.24 | 68.30 | 67.76 | 51.00 (**3.1%**↑) |
| w/ Direct Synthesizing | Geo3K | 26.88 | 43.53 | 69.90 | 68.97 | 52.32 (**5.8%**↑) |
| w/ Original Questions | GeoQA | 27.07 | 43.68 | 68.90 | 68.22 | 51.97 (**5.1%**↑) |
| w/ In-Context Synthesizing | GeoQA | 26.09 | 42.87 | 70.60 | 69.25 | 52.20 (**5.5%**↑) |
| w/ Direct Synthesizing | GeoQA | 26.25 | 44.64 | 71.50 | 68.28 | 52.67 (**6.5%**↑) |
| w/ Original Questions | MMR1 | 26.15 | 44.87 | 72.90 | 68.74 | 53.17 (**7.5%**↑) |
| w/ In-Context Synthesizing | MMR1 | 26.15 | 45.10 | 71.90 | 68.62 | 52.94 (**7.0%**↑) |
| w/ Direct Synthesizing | MMR1 | 26.15 | 44.11 | 70.40 | 67.99 | 52.16 (**5.4%**↑) |

## 4.2 Scenario 1: Unsupervised Training on Standard Datasets

For our experiments, we firstly employ standard training datasets with masked labels to simulate the first scenario (i.e., using human-created questions without ground-truth answers). We conduct MM-UPT on Geometry3k [25], GeoQA [4], and MMR1 [20] using the Qwen2.5-VL-7B [1] model. These datasets cover a diverse set of visual math problems, including geometric diagrams, charts, and structured question formats (multiple-choice and fill-in-the-blank), serving as a strong foundation for models to self-improve the multi-modal mathematical reasoning abilities. More details of these datasets are introduced in Appendix A.3.

Table 1 presents the main results on four challenging multi-modal mathematical reasoning benchmarks. We observe that MM-UPT achieves consistent improvements in average over the base Qwen2.5-VL-7B model across all datasets, also outperforming other baseline methods such as SRLM, LMSI, Genixer, and STIC. Notably, MM-UPT is able to improve the average score from 49.47 (base model) to 53.17 (with MMR1 dataset), demonstrating its effectiveness in leveraging unlabeled data for self-improvement. In comparison, previous baselines provide only marginal gains or even degrade performance on certain benchmarks, highlighting the limitations of existing methods when applied to already strong models in multi-modal reasoning tasks. Furthermore, we find that MM-UPT is even competitive with supervised post-training methods, such as rejection sampling-based SFT [43] and GRPO [37]. These results underscore the potential of MM-UPT to further exploit the knowledge embedded in multi-modal models for self-improvement. Further experiments on the training dynamics, generalization and adaptability of our method are shown in Appendix B.

## 4.3 Scenario 2: Unsupervised Training on Synthetic Datasets

To further explore the potential of MM-UPT, we investigate the use of unlabeled *synthetic data* (mentioned in Section 3.3) to improve MLLMs. This aligns with the ultimate goal of MM-UPT: enabling continual self-improvement even after human-created data is exhausted.

In our experiment, we use the previous two methods in Section 3.3 to generate the synthetic data, leveraging Geometry3K [25], GeoQA [4], and MMR1 [20] as the seed datasets, and Qwen2.5-VL-7B as the base MLLM for data synthesis. MM-UPT is then applied to the same base model (i.e., Qwen2.5-VL-7B) using each of these different synthetic datasets separately. Table 2 presents experimental results using different synthetic data generation strategies. Both in-context and direct synthesizing lead to significant improvements over the base model, achieving performance comparable to training on original human-written questions. This shows that synthetic questions can effectively enhance the model's reasoning ability under MM-UPT. Notably, direct synthesizing even surpasses human-written questions (when applied to Geometry3K and GeoQA) on average, demonstrating the strong ability of the model to generate high-quality textual questions solely based on images. This highlights the

potential for scalable and fully autonomous self-improvement in multi-modal domain via visual-centric data synthesis.

Moreover, we manually examine some synthetic data. We observe that in-context synthesizing often produces questions similar to the original ones by substituting conditions or expressions, resembling data rephrasing. In contrast, direct synthesizing generates more diverse and novel questions. While some of the directly synthesized questions still contain hallucinations, many are of high quality and beneficial for unsupervised post-training. This underscores the potential of the direct synthesizing approach as a simple yet effective method for data generation, without the need for textual in-context examples. Below, we present two illustrative examples that showcase the effectiveness and quality of synthetic questions generated through both approaches.

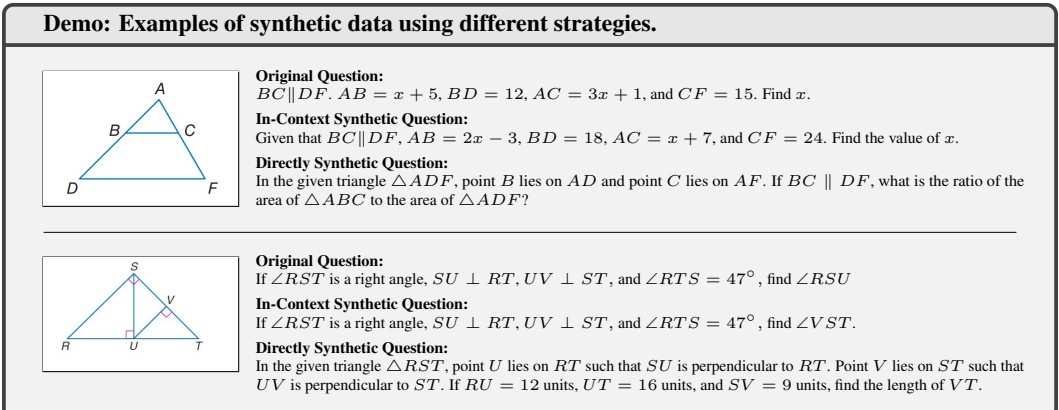

**Demo: Examples of synthetic data using different strategies.**

**Original Question:**
$BC \| DF$. $AB = x + 5$, $BD = 12$, $AC = 3x + 1$, and $CF = 15$. Find $x$.

**In-Context Synthetic Question:**
Given that $BC \| DF$, $AB = 2x - 3$, $BD = 18$, $AC = x + 7$, and $CF = 24$. Find the value of $x$.

**Directly Synthetic Question:**
In the given triangle $\triangle ADF$, point $B$ lies on $AD$ and point $C$ lies on $AF$. If $BC \parallel DF$, what is the ratio of the area of $\triangle ABC$ to the area of $\triangle ADF$?

**Original Question:**
If $\angle RST$ is a right angle, $SU \perp RT$, $UV \perp ST$, and $\angle RTS = 47°$, find $\angle RSU$

**In-Context Synthetic Question:**
If $\angle RST$ is a right angle, $SU \perp RT$, $UV \perp ST$, and $\angle RTS = 47°$, find $\angle VST$.

**Directly Synthetic Question:**
In the given triangle $\triangle RST$, point $U$ lies on $RT$ such that $SU$ is perpendicular to $RT$. Point $V$ lies on $ST$ such that $UV$ is perpendicular to $ST$. If $RU = 12$ units, $UT = 16$ units, and $SV = 9$ units, find the length of $VT$.

## 4.4 Ablation Study

To evaluate the generality and effectiveness of MM-UPT, we conduct an ablation study across a range of backbone models beyond the primary Qwen2.5-VL-7B [1]. Specifically, we apply MM-UPT to several state-of-the-art models of varying scales, including Qwen2.5-VL-3B [1], MM-Eureka-7B [31], and ThinkLite-VL-7B [48]. All models are post-trained using MM-UPT on the Geometry3K dataset [25], without access to any labels (i.e., Scenario 1). As summarized in Table 3, MM-UPT consistently improves the performance of all tested models on average, despite the absence of supervision during post-training. Notably, ThinkLite-VL-7B with MM-UPT achieves the highest average score (54.07), and shows substantial gains on the MathVista [27] benchmark, reaching a score of 74.70. In addition, Qwen2.5-VL-3B, the smallest model in our study, also benefits well from MM-UPT (+7.4% on average), demonstrating the robustness and adaptability of MM-UPT for performance enhancement. These results collectively reveal that MM-UPT can be easily applied to various multi-modal models to enable consistent self-improvement.

Moreover, our results also show that MM-UPT is compatible with supervised GRPO. For instance, MM-Eureka-7B was already tuned with supervised GRPO on the K12 dataset, yet applying MM-UPT on a new unlabeled dataset (Geometry3K) further improved its average score from 53.10 to 53.78 (Table 3). Similarly, ThinkLite-VL-7B was trained with supervised GRPO on the ThinkLite dataset, and applying MM-UPT again boosted its average score to 54.07, including a substantial gain on MathVista (74.70). These results highlight the practical value of MM-UPT as a lightweight refinement step that remains effective even for models already optimized with supervised GRPO, enabling them to leverage new unlabeled data for further improvement.

## 5 Deeper Analysis

Going beyond standard benchmarking, we conduct a deeper analysis to investigate MM-UPT's performance boundaries (Section 5.1 and Section 5.2) and tradeoffs (Section 5.3). This helps better understand its behavior and potential applications.

Table 3: Ablation study using different models besides Qwen2.5-VL-7B. We conduct this experiment on Geometry3K [25] dataset without labels.

| Models | MathVision | MathVerse | MathVista | We-Math | Avg |
|---|---|---|---|---|---|
| Qwen2.5-VL-7B | 24.87 | 43.83 | 66.30 | 62.87 | 49.47 |
| Qwen2.5-VL-7B + MM-UPT | 27.33 | 42.46 | 68.50 | 66.61 | 51.23 (**3.6%**↑) |
| MM-Eureka-7B | 28.06 | 50.46 | 69.40 | 64.48 | 53.10 |
| MM-Eureka-7B + MM-UPT | 28.95 | 50.63 | 69.10 | 66.44 | 53.78 (**1.3%**↑) |
| ThinkLite-VL-7B | 26.94 | 46.58 | 69.00 | 67.99 | 52.63 |
| ThinkLite-VL-7B + MM-UPT | 26.91 | 47.26 | 74.70 | 67.41 | 54.07 (**2.8%**↑) |
| Qwen2.5-VL-3B | 19.47 | 33.58 | 56.30 | 50.63 | 39.00 |
| Qwen2.5-VL-3B + MM-UPT | 22.17 | 32.39 | 57.10 | 55.22 | 41.72 (**7.4%**↑) |

Table 4: Performance of MM-UPT on the difficult ThinkLite-11K dataset. Results show that MM-UPT leads to a decrease in performance when applied to a dataset where the model has limited prior knowledge, highlighting the limitations of majority voting in such scenarios.

| Models | Training Data | MathVision | MathVerse | MathVista | We-Math | Avg |
|---|---|---|---|---|---|---|
| Qwen2.5-VL-7B | – | 24.87 | 43.83 | 66.30 | 62.87 | 49.47 |
| Qwen2.5-VL-7B + MM-UPT | ThinkLite-11K | 21.12 | 37.10 | 59.20 | 59.02 | 44.11 |

## 5.1 Why Does MM-UPT Work?

Majority voting [49] is a fundamental ensemble technique that enhances prediction reliability by aggregating multiple independent responses. In our framework, it offers a simple yet powerful pseudo-reward signal to help model self-improve, particularly when the model are moderately reliable on the unlabeled datasets. We consider a simplified explanation for it using a classical toy example. Suppose that each response hits the correct answer with probability $p > 0.5$ in a binary question. Then, we sample the model's response $n$ times independently. The final answer is determined by a majority vote, that is, the answer that appears more than $n/2$ times. Let $X$ denote the number of correct predictions among the $n$ samples. Since each prediction is correct with probability $p$, $X$ follows a binomial distribution: $X \sim \text{Binomial}(n, p)$. The majority vote is correct if $X > n/2$, and the corresponding probability of this event (denoted as $E$) is:

$$P(E) = \sum_{i=\lceil n/2 \rceil}^{n} \binom{n}{i} p^i (1-p)^{n-i}.$$

When $p > 0.5$, it follows that $P(E) > p$, which means that the ensemble outperforms each individual response. For instance, if $p = 0.7$ and $n = 10$, then $P(E) \approx 0.849$, demonstrating a significant gain over the base accuracy. This analysis reveals the rationality of majority voting to serve as the pseudo-label for deriving reliable reward signal in the unsupervised setting. In our experimental setting, we mainly target datasets that are not especially hard, such as Geometry3k [25], GeoQA [4], and MMR1 [20], for unsupervised post-training. Hence, we hypothesize that the model has a relatively high chance of answering questions in these datasets correctly. This allows the model to yield stable improvements through MM-UPT using majority voting as the pseudo-label.

## 5.2 When Might MM-UPT Fail?

According to the analysis in Section 5.1, it reveals that the effectiveness of MM-UPT diminishes when the model lacks sufficient prior knowledge of the target dataset. To show that, we apply MM-UPT to ThinkLite-11K [48] dataset using Qwen2.5-VL-7B [1]. ThinkLite-11K is collected via difficulty-aware sampling that only retains samples that the model rarely answers correctly. Thus, this setting reflects a scenario where the model is more likely to be wrong than right. In such cases, majority voting amplifies incorrect answers rather than filtering them, leading to degraded performance. As shown in Table 4, applying MM-UPT to ThinkLite-11K results in a significant drop in accuracy across all benchmarks. This suggests that majority voting fails to provide reliable reward signals when

Table 5: Comparison of `pass@10` across different benchmarks. Results show that MM-UPT improves single-response accuracy (`pass@1`, Table 1) but reduces response diversity, leading to a drop in `pass@10`. Supervised GRPO alleviates this issue to some extent.

| Model | MathVision | MathVerse | MathVista | We-Math | Avg |
|---|---|---|---|---|---|
| Qwen2.5-VL-7B | 0.6556 | 0.7307 | 0.8730 | 0.9420 | 0.8003 |
| Qwen2.5-VL-7B + MM-UPT | 0.5661 | 0.6477 | 0.8240 | 0.8621 | 0.7250 |
| Qwen2.5-VL-7B + Supervised GRPO | 0.6164 | 0.6726 | 0.8570 | 0.9075 | 0.7634 |

the model has limited prior understanding of the domain. To address this issue, alternative forms of algorithms using more fine-grained and complex rewarding methods, such as LLM-as-Judge [54, 76] and model collaboration [8, 21], may be necessary. Note that our work represents an initial attempt at self-improvement in MLLMs via GRPO, and we believe that these algorithms are complementary to our approach and could be integrated into our framework in the future.

### 5.3 Trade-Offs in MM-UPT

Although MM-UPT improves `pass@1` accuracy across multiple benchmarks, we also observe a consistent decrease in `pass@n` (for large n, e.g., `pass@10`) performance in Table 5. This reflects a common trade-off between accuracy and diversity in reinforcement learning for multi-modal models. Similar findings have been reported in supervised GRPO [66] (also shown in Table 5), where models tend to collapse onto high-consensus reasoning patterns, thereby reducing output diversity. In MM-UPT, this effect is further amplified because the implicit reward signal from majority voting encourages the model to favor dominant responses, which may suppress minority answers that are occasionally correct. That is said, MM-UPT let the model reinforce problems it can already solve, thereby forgoing the opportunity to tackle more challenging ones and abandoning exploration. While `pass@1` is typically the most relevant metric in real-world applications, mitigating the decline of `pass@n` remains an important open problem for future work.

Another key consideration is the trade-off between training and inference costs. A straightforward alternative to MM-UPT is to apply majority voting directly at inference time, which can also boost accuracy by aggregating multiple outputs. However, inference-time ensembling incurs substantial computational overhead, as $n$ samples must be generated per query, making it expensive and often impractical at scale. By contrast, MM-UPT shifts this cost into a one-time training stage: after refinement, the model produces stronger single-pass outputs, avoiding the need for repeated sampling during deployment. This distinction highlights different usage scenarios: MM-UPT is particularly beneficial when inference efficiency and scalability are critical, whereas inference-time ensembling may be preferable when training resources are limited.

In summary, MM-UPT offers a simple unsupervised post-training framework but also entails clear trade-offs. Balancing accuracy and diversity, and choosing between training-time and inference-time costs, are crucial aspects to consider and represent promising directions for future exploration.

## 6 Conclusion

In this work, we formalize a third-stage post-training paradigm for multi-modal large language models after SFT and RL, termed *Unsupervised Post-Training (UPT)*. We instantiate this paradigm through **MM-UPT**, a simple yet effective framework that leverages majority voting as a self-rewarding mechanism within the GRPO algorithm, guiding models toward consistent and high-confidence responses on multi-modal reasoning tasks. This can be used to further exploit the model's internal knowledge by reinforcing consistent predictions, and thus enables self-improvement without any external supervision. Extensive experiments across multiple benchmarks demonstrate that this method effectively enhances the reasoning performance of strong MLLMs without relying on labeled data or external reward models. Furthermore, we extend the framework to a data self-generation setting, showing that synthetic questions produced by the model itself can further boost performance, revealing a scalable path toward autonomous self-improvement. Future work could explore other fine-grained methods to provide pseudo-reward signals based on our framework, and investigate the scaling laws of unsupervised post-training using synthetic data.

## Acknowledgements and Disclosure of Funding

This project is supported by the National Natural Science Foundation of China (No. 62406192), Opening Project of the State Key Laboratory of General Artificial Intelligence (No. SKLAGI2024OP12), Tencent WeChat Rhino-Bird Focused Research Program, and Doubao LLM Fund.

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

# Appendix

## A Implementation Details

We provide the implementation details of our experiments as follows.

### A.1 Baselines

Here, we explain how we implement different baseline methods in comparison.

**LMSI** [16] employs the majority-voted response as the target for supervised fine-tuning (SFT). For each question, we generate multiple responses and retain the ones that lead to the majority answer for training.

**SRLM** [54] studies Self-Rewarding Language Models, where the model itself is used via LLM-as-a-Judge prompting to provide its own rewards during training. In particular, for each question, we generate multiple candidate responses and use the prompt provided in the original paper to have the MLLM score its own outputs. Among the responses, the one with the highest score is selected as the positive example, and the one with the lowest score as the negative example. These pairs are then used to construct preference datasets for Direct Preference Optimization (DPO) [36].

**Genixer** [74] introduces a comprehensive data generation pipeline consisting of four key steps: (i) instruction data collection, (ii) instruction template design, (iii) empowering MLLMs, and (iv) data generation and filtering. To adapt Genixer in our setting, we remove the first two steps because we already have instruction data. After that, we use Qwen2.5-VL as the backbone model to self-generate responses 16 times per question for each dataset. In the filtering stage, we use the prompt to let the model self-judge the responses following Genixer:

> Here is a question-answer pair. Is $\{Q : X_q, A : X_a\}$ true for this image? Please answer this question with Yes or No.

In addition, Genixer calculates the probability of predicting the "Yes" rather than prompt the model to directly output "Yes" or "No" as the filtering label:

$$P(Y_r|X_I, X_q, X_a) = \prod_i^L p(y_i|X_I, X_q, X_{a,<i}), \tag{3}$$

where $Y_r$ is the predicted judge, $X_I$ is the image, $X_q$ is the question, $X_a$ is the self-generated response, and $L$ is the length the total predicted judge. Then, it proposes a threshold $\lambda$ to control the filtering in the following manner:

$$S^n = \begin{cases} \text{True}, & \text{if } Y_r = \text{Yes and } P(Y_r^n) > \lambda \\ \text{False}, & \text{if } Y_r = \text{Yes and } P(Y_r^n) \leq \lambda \\ \text{False}, & \text{if } Y_r = \text{No} \end{cases} \tag{4}$$

where $S^n$ is the filter label representing keeping or removing the current sample. $P(Y_r^n)$ denotes the probability of the result "Yes" of $n$-th candidates. $\lambda$ is set to 0.7 following the paper.

**STIC** [6] proposes a two-stage self-training algorithm focusing on the image comprehension capability of the MLLMs. In Stage 1, the base MLLM self-constructs its preference dataset for image description using well-designed prompts, poorly-designed prompts, and distorted images with diffusion noise. In Stage 2, a small portion of the previously used SFT data is recycled and infused with model-generated image descriptions to further fine-tune the base MLLM. In particular, since Qwen2.5-VL does not open-source the SFT data, we opt to use the model's self-generated responses sampled from different datasets to represent the previously used SFT data.

### A.2 Benchmarks

We provide some details about the benchmarks we use to evaluate the models' reasoning ability. MathVision [45] is a challenging benchmark containing 3040 mathematical problems with visual contexts from real-world math competitions across 12 grades. It covers 16 subjects over 5 difficulty

levels, including specialized topics like Analytic Geometry, Combinatorial Geometry, and Topology. MathVista [27] is a comprehensive benchmark for evaluating mathematical reasoning in visual contexts. It contains 1000 questions featuring diverse problem types including geometry, charts, and tables. MathVerse [69] is an all-around visual math benchmark designed for an equitable and in-depth evaluation of MLLMs. The test set contains 3940 multi-subject math problems with diagrams from publicly available sources, focusing on Plane Geometry and Solid Geometry. We-Math [35] meticulously collect and categorize 1740 visual math problems in the test set, spanning 67 hierarchical knowledge concepts and 5 layers of knowledge granularity.

For all benchmarks, we prompt the models to place their final answers within a designated box format. We then employ Qwen2.5-32B-Instruct [57] to evaluate answer correctness by comparing the extracted responses with ground truth answers, which often contain complex mathematical expressions. Note that our reported benchmark scores may differ from those in the original papers due to variations in evaluation protocols.

### A.3 Standard Training Datasets

In our experiments, we use three standard training datasets for multi-modal reasoning: Geometry3K [25], GeoQA [4], and MMR1 [20]. Geometry3K consists of 2.1K multiple-choice questions in the training set, covering a wide range of geometric shapes. GeoQA includes 8K fill-in-the-blank questions sourced from the larger Geo170K dataset [10]. MMR1 consists of 7,000 samples and includes both multiple-choice questions and fill-in-the-blank questions. These samples cover a range of tasks, including understanding charts and geometric reasoning.

## B  Additional Experiments

### B.1  Training Dynamics

To better understand the behavior of MM-UPT during training, we monitor several diagnostic metrics, including the majority voting reward and entropy, both of which are label-free and provide insights in the absence of ground-truth supervision. In particular, majority voting reward is calculated following Equation 2. Entropy can be used as an unsupervised objective that measures the uncertainty of the model's generation [46, 52, 68]. For a group of responses $O = \{o_i\}_{i=1}^{G}$ sampled from the question $q$ and image $I$, we cluster the responses according to their meaning. That is, if two responses share the same meaning (i.e., extracted answers), they should be merged into one same cluster in the semantic space. This results to $K(K \leq G)$ clusters $C = \{c_j\}_{j=1}^{K}$. The empirical distribution over clusters is defined as:

$$p(c_j|q, I) = \frac{|c_j|}{G},$$

where $|c_j|$ denotes the number of responses that belongs to $c_j$. The semantic entropy (denoted as $H$) over the model's response meanings distribution can be estimated as follows:

$$H = - \sum_{c_j \in \{C\}} p(c_j|q, I) \log p(c_j|q, I).$$

Figure 2 presents the MM-UPT training curves of the key metrics on Qwen2.5-VL-7B using the MMR1 dataset. We observe that the majority voting reward consistently increases over time, accompanied by a steady decrease in the entropy. This indicates that the model is converging toward more consistent predictions, reflecting improved confidence and stability in its responses.

Additionally, we track the change in average benchmark accuracy and effective rank [52] throughout training. The accuracy exhibits an upward trend, demonstrating that our MM-UPT framework–based on an online reinforcement learning algorithm–effectively enables the model to self-improve continuously and iteratively. The effective rank [52] further measures the amount of knowledge the model comprehends in the datasets. During training, the internal knowledge of the model is exploited, leading to a consistent increase in the effective rank on the benchmark.

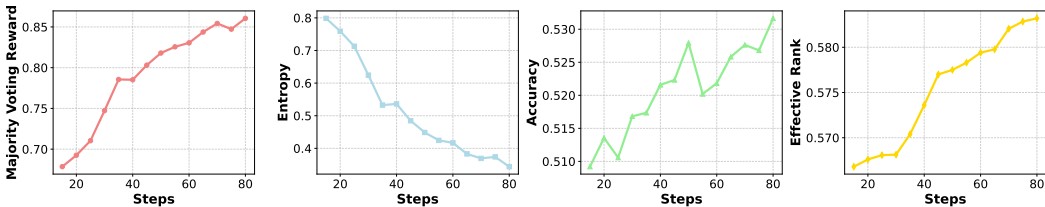

Figure 2: Training dynamics of MM-UPT using Qwen2.5-VL-7B on the MMR1 dataset. We plot the majority voting reward, semantic entropy, and average benchmark accuracy over the course of unsupervised post-training.

Table 6: Performance on non-mathematical VQA benchmarks. We evaluate Qwen2.5-VL-7B before and after applying MM-UPT on the MMR1 dataset. Scores are reported as accuracy.

| Models | ChartQA | IconQA |
|---|---|---|
| Qwen2.5-VL-7B | 71.96 | 54.20 |
| Qwen2.5-VL-7B + MM-UPT | 77.48 (**7.7%**↑) | 56.55 (**4.3%**↑) |

## B.2 Generalization Beyond Multimodal Mathematical Reasoning

A potential concern is that by encouraging convergence toward high-consensus answers, MM-UPT might reduce response diversity and overspecialize on mathematical reasoning, potentially harming performance on other tasks. To investigate whether this method suffers from a negative impact on broader generalization, we extend our evaluation to two non-mathematical visual question answering benchmarks: ChartQA [30], which tests visual perception over charts, and IconQA [26], which focuses on abstract diagram understanding. In particular, we evaluate the Qwen2.5-VL-7B model trained with MM-UPT on the MMR1 dataset. As shown in Table 6, our method not only avoids performance degradation but leads to notable improvements on both benchmarks. This suggests that the underlying mechanism of MM-UPT—reinforcing the generation of more reliable and consistent answers—is a beneficial trait that positively transfers to other domains. The results indicate that the induced consistency does not degrade, and can even enhance, the model's general utility on related visual understanding tasks.

## B.3 Adaptability to Language Tasks

Furthermore, to assess the adaptability of our MM-UPT framework to purely language-based domains, we apply MM-UPT to a language model, Qwen2.5-MATH-7B [58], trained on the DAPO-17K dataset [63] using the same self-rewarding mechanism. We evaluate the resulting model on two popular pure-math benchmarks: MATH [14] and Omni-MATH [9]. As shown in Table 7, MM-UPT leads to substantial improvements on both benchmarks. These results strongly suggest that our framework is not limited to multi-modal settings and can be effectively extended to purely language-based reasoning tasks, provided the base model demonstrates sufficient initial competency.

Table 7: Performance on pure-language mathematical reasoning benchmarks. We evaluate Qwen2.5-MATH-7B before and after applying our unsupervised post-training method.

| Models | MATH | Omni-MATH |
|---|---|---|
| Qwen2.5-MATH-7B | 51.20 | 18.09 |
| Qwen2.5-MATH-7B + UPT | 75.80 (**48.0%**↑) | 32.72 (**80.9%**↑) |

## C Compute Resources

We conduct our experiments using NVIDIA H100-80G and A800-40G GPUs. The experimental time using 8 A800 for training Qwen2.5-VL-7B [1] on the Geometry3K [25] dataset using GRPO is around 10 hours.

