# OpenReview forum: "First SFT, Second RL, Third UPT: Continual Improving Multi-Modal LLM Reasoning via Unsupervised Post-Training"
_NeurIPS.cc/2025/Conference — NeurIPS 2025 poster_

### Official Review · Reviewer_eY5v · 2025-06-26

**Clarity:** 3
**Significance:** 1
**Originality:** 2
**Rating:** 4
**Confidence:** 3

**Summary:**

The paper proposes Multi-Modal Unsupervised Post-Training (MM-UPT), which uses a binary reward signal based on majority voting instead of the usual ground-truth answer in RLVR. This new reward signal is then applied using the GRPO algorithm.

**Questions:**

1. I saw in Figure 2 that the training is only shown for 80 steps. Would the proposed reward scale well over more training steps compared to the baseline with a ground-truth reward?

2. What is the relationship between "training episodes" (line 173) and "steps"? How many rollout is done per training step?

3. Would dropping the temperature to 0 or using majority voting during inference be sufficient to achieve the same performance increase?

4. Would pass@n decrease significantly as every response converges to the majority answer? Such trade-off should be clearly explained in the paper.

**Ethical Concerns:**

["NO or VERY MINOR ethics concerns only"]

**Final Justification:**

I recommend borderline acceptance. The main concern lies in the insufficient analysis presented in the paper—the writing tends to be biased:

- A model with majority voting already achieves similar performance to the fine-tuned model with a majority voting reward. This trade-off between inference cost and training cost should be discussed in the main paper. Which is more important depends on the use case and should be left to the user's judgment.
- The sacrifice in pass@n is not adequately explored, particularly in comparison to the baseline. I do not believe that simply stating the result is consistent with prior literature is sufficient.

That said, given that the main results are solid, I still recommend marginal acceptance—assuming that the above issues will be addressed. A paper should provide a balanced evaluation of a method—not read like a sales pitch that highlights only its strengths.

**Limitations:**

The authors have addressed the limitations and potential negative societal impact of their work.

**Quality:**

2

**Strengths And Weaknesses:**

Strengths

- The proposed learning rule is simple, consisting of a change to the reward function.
- The experimental results are promising, showing that the new method is on par with standard GRPO that uses ground-truth answers.

Weakness

- I am concerned with the scalability of the method across different training data and longer training durations. I understand that the method is essentially decreasing the policy's entropy in a specific way—by steering all responses toward the same answer from majority voting. The positive results in the experiments may not scale as the training data size increases or over more training steps. For instance, standard GRPO with ground-truth answers can still learn from incorrect answers across diverse questions. In contrast, with majority voting, the network might learn to decrease its entropy to an extent that it achieves high rewards on every prompt simply by being more deterministic, rather than by being more correct.
- The paper has insufficient empirical analysis. The theoretical analysis in Section 5.2 is somewhat obvious. More detailed empirical analysis is needed—for example, how do pass@n and majority-voting@n change during training? What was the temperature used in the experiments and evaluation, and how does temperature affect the results?

---

> ### Author Rebuttal · Authors · 2025-07-31
>
> Thank you for your review and detailed feedback. We would like to address your concerns below.
>
> ---
>
> **Q1**. I am concerned with the scalability of the method across different training data and longer training durations. I understand that the method is essentially decreasing the policy's entropy in a specific way—by steering all responses toward the same answer from majority voting. The positive results in the experiments may not scale as the training data size increases or over more training steps. For instance, standard GRPO with ground-truth answers can still learn from incorrect answers across diverse questions. In contrast, with majority voting, the network might learn to decrease its entropy to an extent that it achieves high rewards on every prompt simply by being more deterministic, rather than by being more correct.
>
> **A1**: Thanks for your questions. Firstly, we would like to clarify that **entropy reduction is a common outcome in RL, including in supervised GRPO and other RL methods [1]**. The key is whether convergence improves correctness or merely confidence.
>
> To investigate this, we show that majority voting can offer a rational and effective pseudo-label as analyzed in Section 5.2. Our results in Table 1 and Figure 2 also show that **MM-UPT improves both accuracy and consistency, not just determinism**. Additionally, scaling more training steps (see A3) further reveals that the performance gains can plateau in an expected way similar to supervised GRPO, with no evidence of overconfidence.
>
> Besides, we conduct additional experiments to study the performance when data scales. As shown below, **MM-UPT steadily improves with more unlabeled data, suggesting its scalability within the tested range**.
>
> |Data Size|MathVision|MathVerse|MathVista|We-Math|Avg|
> |-|-|-|-|-|-|
> |0K|0.2487|0.4383|0.6630|0.6287|0.4947|
> |2K|0.2740|0.4459|0.6970|0.6730|0.5225|
> |4K|0.2733|0.4467|0.7010|0.6816|0.5257|
> |6K|0.2615|0.4487|0.7290|0.6874|0.5317|
> |8K|0.2763|0.4490|0.7120|0.6989|0.5341|
>
> We think that fully investigating the scalability of our method is an another important and challenging problem, requiring large-scale computation resources. We are actively working on this direction as part of our future work.
>
> ---
>
> **Q2**. The paper has insufficient empirical analysis. The theoretical analysis in Section 5.2 is somewhat obvious. More detailed empirical analysis is needed—for example, how do pass@n and majority-voting@n change during training? What was the temperature used in the experiments and evaluation, and how does temperature affect the results?
>
> **A2**: Thank you for the questions. We would like to clarify that Section 5.2 serves as an easy-understanding and necessary justification for the rationality of reward design in MM-UPT. Beyond that, we provide sufficient empirical analysis across different benchmarks, models, datasets, and ablations. These collectively form a comprehensive empirical foundation. Moreover, **we also conduct the following study regarding the behavior of pass@n and majority-voting@n (maj@n) for different training steps, and the impact of different temperatures in training**.
>
> As shown below, we observe that maj@10 increases and stabilizes during training, indicating that MM-UPT helps the model consolidate its reasoning, and make more consistent and correct predictions. That said, majority voting benefits are baked into the model. In addition, pass@n tends to increase when n is small (e.g., n=1, shown in Figure 2 in our paper) while decline when n is large (e.g., n=10 shown below). This is a well-known tradeoff in RL training. More details are shown in A5 and A6.
>
> |maj@10|0|10|20|30|40|50|60|70|80|90|
> |-|-|-|-|-|-|-|-|-|-|-|
> |MathVista|0.7050|0.7000|0.7230|0.7280|0.7420|0.7260|0.7410|0.7240|0.7310|0.7210|
> |We-Math|0.6948|0.7098|0.7103|0.7184|0.7224|0.7126|0.7069|0.7080|0.7167|0.7075|
>
> |pass@10|0|10|20|30|40|50|60|70|80|90|
> |-|-|-|-|-|-|-|-|-|-|-|
> |MathVista|0.8730|0.8680|0.8670|0.8630|0.8590|0.8590|0.8480|0.8270|0.8240|0.8150|
> |We-Math|0.9420|0.9178|0.9126|0.9086|0.9000|0.8902|0.8753|0.8631|0.8621|0.8420|
>
> In our training, we use a rollout temperature of 0.7. This strikes a good balance: lower temperatures produce low-diversity outputs, while higher temperatures often lead to lower-quality outputs. We further conduct an ablation on the MMR1 dataset. As shown below, **temperature = 0.7 outperforms both higher and lower settings**, confirming that it helps generate effective pseudo-reward.
>
> |Different rollout temperatures for training|We-Math| MathVista|MathVerse|MathVision|**Avg**|
> |-|-|-|-|-|-|
> |temperature=1.0|0.6851|0.7060|0.4431|0.2763|0.5276|
> |temperature=0.4| 0.6523 |0.6850|0.4338|0.2641|0.5088|
> |temperature=0.7|0.6874|0.7290|0.4487|0.2615|**0.5317**|
>
> ---
>
> **Q3**. I saw in Figure 2 that the training is only shown for 80 steps. Would the proposed reward scale well over more training steps compared to the baseline with a ground-truth reward?
>
> **A3**: Thank you for your question. Our primary goal in this setting was to confirm that the proposed unsupervised method can indeed lead to accuracy gains over steps. To further verify scalability, we extend training to 110 steps. **As the table shown below, MM-UPT continues to improve slightly and eventually converges, with performance plateauing around step 90. This convergence behavior is expected when given a fixed model and dataset, and it also aligns with the supervised GRPO baseline which converges at a similar pace (around step 100)**. While we do not aim to outperform supervised GRPO in absolute terms, our results show that MM-UPT sustains effective training and demonstrates comparable convergence behavior to supervised GRPO.
>
> |Steps|80|85|90|95|100|105|110|
> |-|-|-|-|-|-|-|-|
> |MM-UPT|0.5317|0.5250|**0.5325**|0.5324|0.5259|0.5263  |0.5250|
> |Supervised GRPO|0.5329|0.5344|0.5330|0.5354| **0.5358**|0.5341|0.5239|
>
> ---
>
> **Q4**. What is the relationship between "training episodes" and "steps"? How many rollout is done per training step?
>
> **A4**: Thank you for raising this clarification point. In our implementation, a training episode is a full pass over the unlabeled dataset. Within each episode, we perform one rollout group (G=10) per data point. These groups are used to compute pseudo-rewards and perform a single policy update step per batch. Therefore, **the total number of training steps equals the number of batches per episode times the number of episodes (e.g., 15 episodes × batches per episode)**. Besides, each training step processes a batch of examples sampled from the training dataset. In our experiments, the batch size is set to 128, so each training step involves **128 × 10 = 1280** rollouts.
>
> ---
>
> **Q5**. Would dropping the temperature to 0 or using majority voting during inference be sufficient to achieve the same performance increase?
>
> **A5**: Thanks for your question and we conduct these follow-up experiments to help explain.
>
> Firstly, our results show that temperature=0 inference (using Qwen2.5VL, i.e., the base model) alone fails to match MM-UPT. This is because **temperature=0 yields a single, deterministic output that may reflect model biases or local optima, while majority voting aggregates multiple diverse samples to reduce randomness and improve robustness** [4].
>
> Besides, we find that **majority voting during inference does bring performance improvements, but it requires sampling multiple outputs per query, which is computationally expensive and impractical at scale**. Importantly, **MM-UPT allows us to achieve strong performance on pass@1, meaning that the majority voting benefits are baked into the model itself, without needing to rely on expensive sampling-based inference**, making the model more efficient and practical for deployment. Furthermore, we observe that applying majority voting at inference after MM-UPT training can lead to better performance, suggesting that MM-UPT improves both the model’s reasoning consistency and accuracy.
>
> | |We-Math|MathVista|MathVerse|MathVision|**Avg**|
> |-|-------|---------|---------|----------|-|
> |Base (pass@1 with temprature=0)|0.6471|0.6740|0.4353|0.2573|0.5034|
> |+ MM-UPT (pass@1)|0.6874|0.7290|0.4487|0.2615|0.5317|
> |Base (maj@10)|0.6948|0.7050|0.4802|0.2832|0.5408|
> |+ MM-UPT (maj@10)|0.7075|0.7310|0.4772|0.2836|0.5498|
>
> ---
>
> **Q6**. Would pass@n decrease significantly as every response converges to the majority answer? Such trade-off should be clearly explained in the paper.
>
> **A6**: Thank you for the question. Encouraging consistency can indeed reduce diversity, lowering pass@n for large n. **This is a well-known trade-off in many RL methods no matter supervised or not [2,3]**. For example, [2] observes that while RL-trained models outperform their base models on pass@n at smaller values of n (e.g., n=1), base models achieve higher pass@n score when n becomes larger.
>
> To further investigate this, we evaluate pass@10 (meaning a larger n) in our case. As shown below, MM-UPT leads to a modest drop in pass@10 compared to the base model. However, a similar drop is also observed with supervised GRPO, suggesting that this is a general consequence of RL training, rather than a specific flaw in MM-UPT. Importantly, **MM-UPT improves pass@1 significantly (i.e., at smaller values of n) according to Table 1 in our paper, showing that the model is becoming more accurate, not merely more deterministic**.
>
> |pass@10|We-Math|MathVista|MathVerse|MathVision|**Avg**|
> |-|-|-|-|-|-|
> |Base|0.9420|0.8730|0.7307|0.6556|0.8003|
> |+ MM-UPT|0.8621|0.8240|0.6477|0.5661|0.7250|
> |+ Supervised GRPO|0.9075|0.8570|0.6726|0.6164|0.7634|
>
> ---
>
> [1] Cui et al. "The entropy mechanism of reinforcement learning for reasoning language models."
>
> [2] Yue et al. "Does reinforcement learning really incentivize reasoning capacity in llms beyond the base model?"
>
> [3] Dang et al. "Assessing diversity collapse in reasoning."
>
> [4] Wang et al. "Self-consistency improves chain of thought reasoning in language models."

---

> > ### Comment · Reviewer_eY5v · 2025-08-04
> >
> > Thanks for the response. Please see my comment below:
> >
> > Q1
> >
> > > Firstly, we would like to clarify that entropy reduction is a common outcome in RL, including in supervised GRPO and other RL methods [1].
> >
> > I did not state whether entropy reduction is or is not a common outcome in RL. My question pointed out that the method reduces entropy in a specific way, and I was asking whether this specific approach scales well with training data size. The table suggests that it does scale with dataset size, which addresses my question.
> >
> > Q2 & Q6
> >
> > > As shown below, we observe that maj@10 increases and stabilizes during training, indicating that MM-UPT helps the model consolidate its reasoning, and make more consistent and correct predictions. That said, majority voting benefits are baked into the model. In addition, pass@n tends to increase when n is small (e.g., n=1, shown in Figure 2 in our paper) while decline when n is large (e.g., n=10 shown below). This is a well-known tradeoff in RL training. More details are shown in A5 and A6.
> >
> > I am unable to locate the mentioned A5 and A6.
> >
> > I assume the values 0 to 90 in the two tables represent training steps. maj@10 appears to remain relatively stable during training, while pass@10 decreases significantly. I do not believe this significant drop in pass@10 can be dismissed simply by stating, “This is a well-known tradeoff in RL training.” In fact, both con@64 and pass@8 increase steadily in standard RLVR for LLMs [1,2]. Figure 1 in [3] also shows that pass@8 improves steadily with training steps. This suggests a unique phenomenon associated with using majority-voting rewards, which should be discussed in more detail.
> >
> > I believe the drop in pass@n is unsurprising when using majority-voting rewards compared to supervised rewards. Under majority-voting, minority responses—even if correct—tend to be suppressed with prolonged training. The key is to address this phenomenon objectively, rather than dismissing it as a common occurrence in RL training.
> >
> > My understanding on common RLVR with supervised reward is that for very large n, RLVR does not improve pass@n and may even lead to a slight drop. However, for smaller n values such as 8 or 10, we typically observe steady improvement throughout training—though the effect is less pronounced than for pass@1. This is consistent with results in [1,3]. Therefore, I am particularly interested in understanding why the table in Q6 shows a drop in pass@10 for GRPO.
> >
> > Q5
> >
> > From the table, it seems that majority voting applied to the base model already achieves performance similar to the RL-trained model using majority-voting rewards. This represents a tradeoff between computational cost during training and inference. The RL-trained model requires significantly more time to train, but less time during inference due to the absence of majority voting. Again, this tradeoff should be discussed objectively in the paper, rather than emphasizing only one side.
> >
> >
> > [1] Guo, D., Yang, D., Zhang, H., Song, J., Zhang, R., Xu, R., ... & He, Y. (2025). Deepseek-r1: Incentivizing reasoning capability in llms via reinforcement learning. arXiv preprint arXiv:2501.12948.
> >
> > [2] Zeng, W., Huang, Y., Liu, Q., Liu, W., He, K., Ma, Z., & He, J. Simplerl-zoo: Investigating and taming zero reinforcement learning for open base models in the wild, 2025. URL https://arxiv. org/abs/2503.18892.
> >
> > [3] Yue, Y., Chen, Z., Lu, R., Zhao, A., Wang, Z., Song, S., & Huang, G. (2025). Does reinforcement learning really incentivize reasoning capacity in llms beyond the base model?. arXiv preprint arXiv:2504.13837.

---

> ### Author Response · Authors · 2025-08-04
> **Response to Reviewer eY5v (1)**
>
> Thank you sincerely for the active discussion. We would like to respond to the points raised in your follow-up questions.
>
> ---
>
> **Q7**: I did not state whether entropy reduction is or is not a common outcome in RL. My question pointed out that the method reduces entropy in a specific way, and I was asking whether this specific approach scales well with training data size. The table suggests that it does scale with dataset size, which addresses my question.
>
> **A7**: Thank you for the clarification. We appreciate your emphasis on the specific mechanism of entropy reduction in MM-UPT through majority voting. We also appreciate that the additional experiments we provide are helpful and address your question. Thank you again for pointing this out.
>
> ---
>
> **Q8**: I am unable to locate the mentioned A5 and A6.
>
> **A8**: Thanks for your attention to this detail. A5 and A6 correspond to our responses to Q5 and Q6. Due to space constraints in the initial rebuttal, we were unable to elaborate further at that time, and instead included the answers in continuous text.
>
> ---
>
> **Q9**: I assume the values 0 to 90 in the two tables represent training steps. maj@10 appears to remain relatively stable during training, while pass@10 decreases significantly. I do not believe this significant drop in pass@10 can be dismissed simply by stating, “This is a well-known tradeoff in RL training.” In fact, both con@64 and pass@8 increase steadily in standard RLVR for LLMs [1,2]. Figure 1 in [3] also shows that pass@8 improves steadily with training steps. This suggests a unique phenomenon associated with using majority-voting rewards, which should be discussed in more detail.
>
> **A9**: Thank you for the detailed follow-up. The values 0 to 90 in the two tables indeed represent training steps. We appreciate your time in verifying these references focusing **RLVR for language models** [1,2,3]. You are correct that Figure 1 in [3] you observe does show gains in pass@8 for language-only models trained on math dataset. However, we encourage double-checking **Figure 4 in [3]** to confirm that this pattern is consistent with our findings in the **multi-modal setting**. In particular, **this paper [3] also evaluates the same multi-modal model (Qwen2.5-VL) as we use**. **In Figure 4 of [3], it is shown that for multi-modal model, RLVR with supervised rewards starts to underperform the base model on pass@n when n > 4**. This aligns well with our observations. It also suggests that the drop in pass@10 for multi-modal models is indeed not a unique phenomenon associated with using majority-voting rewards, but rather a broader challenge for RLVR training of multi-modal models.
>
> ---
>
> **Q10**: I believe the drop in pass@n is unsurprising when using majority-voting rewards compared to supervised rewards. Under majority-voting, minority responses—even if correct—tend to be suppressed with prolonged training. The key is to address this phenomenon objectively, rather than dismissing it as a common occurrence in RL training.
>
> **A10**: Thank you for highlighting this important point.
>
> From a practical standpoint, we would like to clarify that our work focuses on improving the multi-modal model's pass@1 without any external supervision in the post-training, and thus we only evaluate pass@1 in our initial submission. **This metric, pass@1, is more aligned with real-world application scenarios than pass@n (for large n), where users typically expect a single, reliable response to their query without a known ground-truth label**. Our MM-UPT indeed improves pass@1 substantially, reflecting a meaningful enhancement in single-attempt response quality.
>
> In addition, regarding the drop in pass@n, your observation is accurate and it is a direct consequence of the majority-voting reward mechanism. This is expected given the weaker and noisier nature of unsupervised rewards. In supervised GRPO, the model receives ground-truth signals, which allow it to retain useful minority responses. In contrast, MM-UPT uses majority voting to approximate pseudo-labels. As a result, correct but minority answers are more likely to be penalized as training progresses. This naturally leads to a stronger convergence toward majority modes, and hence a more noticeable drop in pass@10. **We will add this detailed discussion of the tradeoff in our revised version**.
>
> Moreover, **we agree that mitigating the drop in pass@n (for large n) is an important direction for the RLVR research field. Addressing this issue requires more sophisticated reward aggregation strategies for RLVR, which is beyond the scope of our current work's focus on self-improvement.** We view this as a valuable and open research problem, and hope our current findings can help motivate further investigation in this area.
>
> Thanks again for your feedback. We will include a detailed and objective discussion of this trade-off in the revised version of our paper.

---

> ### Author Response · Authors · 2025-08-04
> **Response to Reviewer eY5v (2)**
>
> **Q11**: My understanding on common RLVR with supervised reward is that for very large n, RLVR does not improve pass@n and may even lead to a slight drop. However, for smaller n values such as 8 or 10, we typically observe steady improvement throughout training—though the effect is less pronounced than for pass@1. This is consistent with results in [1,3]. Therefore, I am particularly interested in understanding why the table in Q6 shows a drop in pass@10 for GRPO.
>
> **A11**: Thank you for raising this nuanced point. Indeed, your understanding aligns well with many observations reported in the literature for language-only models. However, **we would like to clarify that our experiments are conducted on a multi-modal setting, where the training dynamics can differ significantly from unimodal (language-only) setups**. This distinction is critical when interpreting trends such as pass@n under RLVR.
>
> As we have also discussed in A9 (for Q9), **our findings are consistent with the multi-modal experiments reported in Figure 4 of [3]**. That figure shows that for multi-modal models like Qwen2.5-VL, RLVR with supervised rewards improves pass@1 but leads to a drop in pass@n for n > 4 as training progresses. This aligns with our observations in both the MM-UPT and supervised GRPO settings. Therefore, the decline in pass@10 in our Table in Q6 is not unexpected, and is in line with existing multi-modal RLVR results.
>
> The discrepancy between unimodal and multi-modal results may be due to several factors, including differences in input modality fusion, as well as the complexity of training datasets and evaluation benchmarks. As a result, multi-modal models may be more susceptible to converging towards consistent answer modes, causing diversity loss at a lower value of n (in pass@n) compared to unimodal LLMs.
>
> We will clarify this distinction in the revised paper to avoid confusion, and we thank you again for pointing out this important difference.
>
> ---
>
> **Q12**: From the table, it seems that majority voting applied to the base model already achieves performance similar to the RL-trained model using majority-voting rewards. This represents a tradeoff between computational cost during training and inference. The RL-trained model requires significantly more time to train, but less time during inference due to the absence of majority voting. Again, this tradeoff should be discussed objectively in the paper, rather than emphasizing only one side.
>
>
> **A12**: Thanks for your suggestion. We indeed agree that a more objective discussion of the trade-off between training and inference cost is necessary. In particular, we believe that majority voting at inference offers a lightweight alternative in some use cases, while MM-UPT offers performance improvements under the pass@1 setting without requiring sampling at inference, which is more appealing for large-scale deployment. This is because **the cost of LLM inference when deployed at scale for real-world scenarios often exceeds that of post-training (at once)**. Therefore, reducing inference-time overhead by avoiding sampling-based majority voting is not only a technical advantage but also a cost-saving measure in real-world systems. **Our method effectively amortizes the cost of ensembling at inference into a one-time post-training process**, producing a single, improved, and efficient model that is cheaper to run at scale. We will revise the paper to explicitly include this tradeoff.
>
> ---
>
> [1] Guo, D., Yang, D., Zhang, H., Song, J., Zhang, R., Xu, R., ... & He, Y. (2025). Deepseek-r1: Incentivizing reasoning capability in llms via reinforcement learning. arXiv preprint arXiv:2501.12948.
>
> [2] Zeng, W., Huang, Y., Liu, Q., Liu, W., He, K., Ma, Z., & He, J. Simplerl-zoo: Investigating and taming zero reinforcement learning for open base models in the wild, 2025. URL https://arxiv.org/abs/2503.18892.
>
> [3] Yue, Y., Chen, Z., Lu, R., Zhao, A., Wang, Z., Song, S., & Huang, G. (2025). Does reinforcement learning really incentivize reasoning capacity in llms beyond the base model?. arXiv preprint arXiv:2504.13837.

---

> > ### Comment · Area_Chair_nj6y · 2025-08-04
> > **Please respond to the author's rebuttal post**
> >
> > Hi Reviewer eY5v, I see no response letting me know whether or not the rebuttal has changed your opinion. Could you please let me and the authors know by engaging? This process is critical to enabling the (S)ACs to make a decision on this work.
> >
> > --Your AC

---

> > ### Comment · Reviewer_eY5v · 2025-08-05
> >
> > Thanks for the response. My concerns are addressed, and score is updated accordingly.

---

> > > ### Author Response · Authors · 2025-08-05
> > > **Thanks for updating your score.**
> > >
> > > Thank you for your reconsideration of our paper and the adjustment of the score. We assure you that the valuable suggestions and insights from you and other reviewers, as well as our explanations, will certainly be integrated into our revised version. We sincerely appreciate the time and effort you've dedicated to this. Thanks again for your review and comments.

---

### Official Review · Reviewer_XWR7 · 2025-06-29

**Clarity:** 3
**Significance:** 3
**Originality:** 3
**Rating:** 4
**Confidence:** 4

**Summary:**

This paper introduces MM-UPT for unsupervised post-training of MLLMs using the GRPO RL algorithm without any human-annotated supervision. MM-UPT utilizes pseudo label based on majority voting over multiple sampled responses to guide learning. Experiments show that the proposed framework improves the reasoning ability of Qwen2.5-VL-7B and other base models on benchmarks like MathVista and We-Math, outperforming prior unsupervised baselines and even approaching supervised GRPO performance.

**Questions:**

Same as stated in weaknesses. If the authors can answer my questions, I am willing to raise my score.

**Ethical Concerns:**

["NO or VERY MINOR ethics concerns only"]

**Final Justification:**

I have read the rebuttal from authors and raise my score accordingly.

**Limitations:**

yes

**Quality:**

3

**Strengths And Weaknesses:**

**Strengths**:
1. The performance of the proposed method surpasses the previous unsupervised baselines, indicating the potential of unsupervised RL post-training.
2. The analysis about the failure cases and reasons why MM-UPT can work is insightful.
3. The paper is easy to follow with clear structures.

**Weaknesses**:
1. Besides what have been analyzed for MM-UPT, another question is the upper bound of MM-UPT. More specifically, what is the relationship between the amount of the unsupervised data and the performance gain. Can the model benefit from unsupervised training endlessly?
2. For the comparison with supervised baseline like GRPO, I am wondering the epoch setting for supervised GRPO, as after training for more epochs, the sampled responses would be more consistent for supervised GRPO as well. If both of them use the same training data as the seed, will them have similar upper bound after convergence.

---

> ### Author Rebuttal · Authors · 2025-07-31
>
> Thank you for recognizing the strengths of our method and analysis. We would like to address your questions regarding the upper bound of MM-UPT and its relationship with supervised GRPO below.
>
> ---
>
> **Q1**. Besides what have been analyzed for MM-UPT, another question is the upper bound of MM-UPT. More specifically, what is the relationship between the amount of the unsupervised data and the performance gain. Can the model benefit from unsupervised training endlessly?
>
> **A1**: Thank you for this question. We agree that understanding the upper bound of MM-UPT is crucial for assessing its scalability. To explain this, we conduct additional experiments analyzing the relationship between the size of the unsupervised training data and the resulting performance. **The results below show that the performance gradually improves as the amount of unsupervised data increases within the tested range**.
>
> Notably, the general scaling law [1] rules state that under a fixed model size, performance improvements from additional data will eventually saturate. Therefore, **while we have not yet observed an obvious plateau in our current setting, we believe the performance of MM-UPT is indeed upper-bounded by the model’s capacity**. However, this further exploration would require significantly more computational resources and access to much larger, more diverse high-quality unsupervised datasets. Due to the resource constraints, currently we are unable to exhaustively explore this saturation point. We are actively working to scale up both data and training resources to better explore this interesting question.
>
>
> | Data Size | MathVision | MathVerse | MathVista | We-Math | Avg    |
> |:--------- |:---------- |:--------- |:--------- |:------- |:------ |
> | 0K        | 0.2487     | 0.4383    | 0.6630    | 0.6287  | 0.4947 |
> | 2K      | 0.2740     | 0.4459    | 0.6970    | 0.6730  | 0.5225 |
> | 4K      | 0.2733     | 0.4467    | 0.7010    | 0.6816  | 0.5257 |
> | 6K        | 0.2615     | 0.4487    | 0.7290    | 0.6874  | 0.5317 |
> | 8K          |  0.2763          |  0.4490         |     0.7120      |    0.6989     | 0.5341       |
>
>
>
> [1] Kaplan, Jared, et al. "Scaling laws for neural language models."
>
> ---
>
> **Q2**. For the comparison with supervised baseline like GRPO, I am wondering the epoch setting for supervised GRPO, as after training for more epochs, the sampled responses would be more consistent for supervised GRPO as well. If both of them use the same training data as the seed, will them have similar upper bound after convergence.
>
>
> **A2**: Thank you for the question regarding the training settings and the convergence bounds of our method compared to supervised GRPO.
>
>
> Firstly, we would like to clarify that all comparisons were conducted under fair conditions. As stated in Section 4 of our paper, **both our unsupervised MM-UPT and the supervised GRPO baselines were trained using the exact same hyperparameter settings**, including the number of training episodes (15). Both methods have converged by the end of training. Note that in our experiment, **we mask the labels of supervised data to perform unsupervised GRPO, simply to simulate a scenario where annotated labels are unavailable**.
>
>
> Secondly, **if both methods use the same seed data, they will indeed have a similar upper bound**. Practically, a supervised reward signal is superior to the pseudo-reward from majority voting. Therefore, **once a model has been fully trained with supervised GRPO on a given dataset, its performance is already near its peak capacity and the sampled responses would become consistent for that specific data distribution**. Applying MM-UPT subsequently on the same data (with labels masked) would not be expected to yield further significant improvements, as the potential for gains has already been exhausted by the more accurate supervised signal. To empirically validate this, we conduct an additional experiment. The results below show that after training with supervised GRPO on the MMR1 dataset, further applying MM-UPT on the same unlabeled MMR1 data results in a nearly identical performance upper bound. Thus, **in practical application scenarios, if the dataset has human-annotated labels, one can simply use supervised GRPO for training**.
>
> | Training Method | MathVision | MathVerse | MathVista | We-Math | Avg |
> | :--- | :--- | :--- | :--- | :--- | :--- |
> | Supervised GRPO (on MMR1 dataset) | 0.2928 | 0.4690 | 0.6960 | 0.6839 | 0.5354 |
> | Supervised GRPO (on MMR1 dataset) + MM-UPT (on unlabeled MMR1) | 0.2924 | 0.4685 | 0.7000 | 0.6787 | 0.5349 |
>
>
> Inspired by your question, we think that a more practical and important question is whether our unsupervised method is compatible with supervised GRPO. Specifically, **after conducting supervised GRPO on the model using one dataset, can MM-UPT further improve this model using a new, unlabeled dataset?** Actually, our paper has already provided evidence that the answer is **yes**. As shown in Table 3 in our paper (also the table below), our method can further improve the MM-Eureka-7B model. Note that the MM-Eureka-7B model was already tuned with supervised GRPO on the K12 dataset. **By applying MM-UPT with a new, unlabeled dataset (Geometry3k), we are able to further boost MM-Eureka-7B's average performance from 53.10 to 53.78**. This demonstrates the key value of our method: it serves as an effective tool for continual refinement, allowing already powerful models to keep improving by leveraging new and unlabeled data.
>
> | Training Method | MathVision | MathVerse | MathVista | We-Math | Avg |
> | :--- | :--- | :--- | :--- | :--- | :--- |
> | Supervised GRPO (on K12 dataset) | 0.2806 | 0.5046 | 0.6940 | 0.6448 | 0.5310 |
> | Supervised GRPO (on K12 dataset) + MM-UPT (on unlabeled Geometry3k) | 0.2895 | 0.5063 | 0.6910 | 0.6644 | **0.5378** |
>
> In summary, while the performance upper bound is similar when applying both methods to the same dataset, the true strength of MM-UPT lies in its ability to effectively and compatibly refine models using new, unlabeled data, showcasing a promising path for effective self-improvement.

---

> ### Author Response · Authors · 2025-08-04
> **We would be grateful if you could take a look at the response**
>
> Dear Reviewer XWR7:
>
> We sincerely appreciate your valuable time devoted to reviewing our manuscript. We would like to gently remind you of the approaching deadline for the discussion phase. We have diligently addressed the issues you raised in your feedback, providing detailed explanations and experiments. For instance, we have clarified the upper bound of MM-UPT and its relationship with supervised GRPO with further experiments. Would you kindly take a moment to look at it?
>
> We are very enthusiastic about engaging in more in-depth discussions with you.
>
> With warm regards,
>
> Authors

---

> > ### Comment · Reviewer_XWR7 · 2025-08-04
> >
> > Thanks for your effort in rebuttal. It resolves my concerns mostly. I will raise my score accordingly. As you mentioned, I agree that exploring a model already trained by GRPO with labeled datasets is a more important setting. The improvement of Eureka-7B's average performance from 53.10 to 53.78 with unlabeled geometry3k seems not a thorough evaluation with very marginal improvement. Hope you can conduct more solid evaluations on different base models (besides QwenVL) and more unlabeled datasets with the above setting to expand the impact of your work.

---

> > > ### Author Response · Authors · 2025-08-05
> > > **Thanks for raising your score.**
> > >
> > > Thank you for your reconsideration of our paper and the adjustment of the score. We will continue to actively conduct more comprehensive evaluations on a broader range of base models and unlabeled datasets to further validate and expand the impact of our approach. We assure you that the valuable suggestions and insights from you and other reviewers, as well as our explanations, will certainly be integrated into our revised version.
> > >
> > > We sincerely appreciate the time and effort you've dedicated to this. Thanks again for your review and comments.

---

> ### Author Response · Authors · 2025-08-09
> **Additional Experimental Results**
>
> Thank you again for your constructive feedback. Following your suggestion, we expand our experiments to evaluate models already trained with GRPO on labeled datasets. Specifically, we consider two additional settings: (1) a model trained on the labeled K12 dataset and then enhanced with MM-UPT using the unlabeled MMR1 dataset; and (2) a model trained on the labeled ThinkLite dataset and then enhanced with MM-UPT using the unlabeled Geometry3k dataset.
>
> | Training Method             | MathVision | MathVerse | MathVista | We-Math | Avg        |
> | --------------------------- | ---------- | --------- | --------- | ------- | ---------- |
> | Supervised GRPO (on K12 dataset)       | 0.2806     | 0.5046    | 0.6940    | 0.6448  | 0.5310     |
> | Supervised GRPO (on K12 dataset) + MM-UPT (on unlabeled MMR1 dataset)             | 0.2829     | 0.5028    | 0.7180    | 0.6621  | **0.5415** |
> | Supervised GRPO (on ThinkLite dataset) | 0.2694     | 0.4658    | 0.6900    | 0.6799  | 0.5263     |
> | Supervised GRPO (on ThinkLite dataset) + MM-UPT (on unlabeled Geometry3k dataset)       | 0.2691     | 0.4726    | 0.7470    | 0.6741  | **0.5407** |
>
> These results demonstrate that MM-UPT consistently enhance performance across different labeled and unlabeled training datasets with more significant improvements, confirming its generality. We plan to further evaluate our method on additional base models to broaden its impact.

---

### Official Review · Reviewer_nZ3M · 2025-07-02

**Clarity:** 4
**Significance:** 4
**Originality:** 3
**Rating:** 5
**Confidence:** 4

**Summary:**

The paper proposes an unsupervised RL training framework, MM-UPT, to improve the reasoning capability of MLLMs, which uses the majority voting as a proxy for the ground truth answer. The authors conduct extensive experiments and demonstrate the effectiveness of their method. They also provide interesting discussions and analysis

**Questions:**

How does the weight beta of the KL divergence affect the training result?

**Ethical Concerns:**

["NO or VERY MINOR ethics concerns only"]

**Final Justification:**

After reviewing the rebuttals, I keep my original score of 5.

**Quality:**

3

**Strengths And Weaknesses:**

Strengths:
- The topic is very related. The proposed method, MM-UPT, is simple but effective. The ground truth answer is replaced with a majority voted answer, so GRPO-like training can be done in an unsupervised way.
- MM-UPT outperforms prior unsupervised methods and approaches fully‑supervised GRPO on several benchmarks.
- The authors investigate synthetic question generation, and demonstrate that synthetic questions can be useful to improving model reasoning ability. The proposed method works on smaller scale models (3B) too.
- The paper is clearly written, and the authors provide open-sourced code.

Weaknesses:
- If the starting accuracy is lower than 0.5, this method may not work, so it depends on the quality of the pre-trained model. However, I understand this is a basic requirement for almost all unsupervised methods.

---

> ### Author Rebuttal · Authors · 2025-07-31
>
> Thank you for your positive and encouraging feedback, and for recognizing the effectiveness and clarity of our work. We would like to respond to your questions below.
>
> ---
>
> **Q1**. If the starting accuracy is lower than 0.5, this method may not work, so it depends on the quality of the pre-trained model. However, I understand this is a basic requirement for almost all unsupervised methods.
>
> **A1**: Thank you for highlighting this important observation. We agree that MM-UPT requires the base model to exhibit non-trivial initial accuracy (i.e., >0.5) for majority voting to yield meaningful pseudo-rewards. This dependency on pre-trained model quality is indeed a limitation. However, as you rightly notes, it is a fundamental assumption shared by most unsupervised or self-improving frameworks. Without a reasonably competent starting point, self-generated signals are likely to reinforce noise rather than useful knowledge. In this work, we position MM-UPT as a post-training enhancement technique rather than a substitute for pretraining or supervised fine-tuning. MM-UPT is best used to amplify existing knowledge in models that already demonstrate basic competency on the target domain.
>
> ---
>
> **Q2**. How does the weight beta of the KL divergence affect the training result?
>
> **A2**: Thank you for the question. The KL divergence weight $\beta$ plays a critical role in stabilizing MM-UPT training. If $\beta$ is set too small, the policy may deviate too far from the reference model, leading to instability and training collapse. Conversely, setting $\beta$ too large overly constrains the policy, resulting in minimal updates. Following prior works [1], we adopt an empirical value of **1.0e-2** as the default setting. To further validate this choice, **we additionally conduct an ablation study on the KL coefficient using the MMR1 dataset**. As shown below, setting $\beta$ = 1.0e-2 achieves a good balance between stability and learning performance on average.
>
> | KL Coefficient ($\beta$) | We-Math    | MathVista  | MathVerse  | MathVision | **Avg** |
> | ------------------------ | ---------- | ---------- | ---------- | ---------- | ------------------ |
> | 2.0e-2                   | 0.6943     | 0.7160     | 0.4462     | 0.2572     | 0.5284             |
> | 5.0e-3                   | 0.6960     | 0.6920     | 0.4368     | 0.2635     | 0.5221             |
> | **1.0e-2**                   | 0.6874     | 0.7290     | 0.4487     | 0.2615     | **0.5317**             |
>
> [1] Deng, Yihe, et al. "Openvlthinker: An early exploration to complex vision-language reasoning via iterative self-improvement."

---

> > ### Comment · Reviewer_nZ3M · 2025-08-05
> > **Thanks for your reply**
> >
> > Thanks for the author's reply, which has addressed my concerns.

---

> > > ### Author Response · Authors · 2025-08-05
> > >
> > > Thank you for your follow-up and for acknowledging our response. We're glad we were able to address your concerns. We sincerely appreciate the time and effort you've dedicated to this. Thanks again for your review and comments.

---

> ### Comment · Area_Chair_nj6y · 2025-08-04
> **Please respond to the author's rebuttal post**
>
> Hi Reviewer nZ3M, I see no response letting me know whether or not the rebuttal has changed your opinion. Could you please let me and the authors know by engaging? This process is critical to enabling the (S)ACs to make a decision on this work.
>
> --Your AC

---

### Official Review · Reviewer_QjgG · 2025-07-05

**Clarity:** 3
**Significance:** 3
**Originality:** 4
**Rating:** 4
**Confidence:** 4

**Summary:**

This paper introduces MM-UPT, a framework for unsupervised post-training of MLLMs that enables self-improvement without external supervision. The method builds on GRPO by replacing traditional reward signals with a self-rewarding mechanism based on majority voting over multiple sampled responses. Given unlabeled multi-modal data, the model samples multiple responses, determines pseudo-labels through majority voting, and updates itself via GRPO. The authors evaluate MM-UPT on multi-modal mathematical reasoning benchmarks, showing that it significantly improves performance compared to base models and outperforms other unsupervised baselines. They also explore synthetic data generation strategies, finding that model-generated questions can achieve competitive results. The paper provides analysis of when this approach succeeds and when if fails.

**Questions:**

See Weakness

**Ethical Concerns:**

["NO or VERY MINOR ethics concerns only"]

**Final Justification:**

After reviewing the rebuttals, I keep my initial score.

**Limitations:**

Yes

**Quality:**

3

**Strengths And Weaknesses:**

Strengths

- The paper proposes a remarkably simple yet effective approach for unsupervised GRPO based on majority voting, achieving strong results—even outperforming supervised training on some benchmarks.

- It provides a thoughtful analysis of the conditions under which majority-voting-based unsupervised GRPO is likely to succeed—namely, when the training data is relatively easy and the base model already has reasonable initial accuracy. It also clearly acknowledges limitations, such as failure cases on harder datasets where the model lacks sufficient prior knowledge.

Weaknesses

- It seems that the proposed unsupervised GRPO method, MM-UPT, cannot consistently improve performance across all benchmarks. As shown in Table 1, the performance on MathVerse drops modestly in some data settings. Can the authors explain the possible reasons for this inconsistency?

- Based on Table 1, it seems that using different kinds of data brings significantly different improvements on different benchmarks. However, the paper lacks analysis on why different training data improve performance differently and lacks experiments investigating whether combining these datasets could bring additional performance improvements.

- Based on the basic idea of majority voting and the analysis of semantic entropy during training, this method encourages the model to converge toward more consistent predictions. My concern is that this may reduce response diversity. As a result, I'm wondering whether this method affects the model's performance on other non-math benchmarks?

- The method's effectiveness critically depends on the base model's initial accuracy being above chance (>0.5). This is a strong assumption that limits applicability, as the method cannot bootstrap knowledge on genuinely difficult or out-of-domain datasets where the model would initially have low performance.

- Given that evaluation focuses solely on multi-modal mathematical reasoning, can this method transfer to improve pure language math tasks or other non-math tasks?

---

> ### Author Rebuttal · Authors · 2025-07-31
>
> Thank you for highlighting the simplicity and effectiveness of MM-UPT, and for appreciating our analysis of its strengths and limitations. We would like to address your concerns point by point below.
>
> ---
>
> **Q1**. It seems that the proposed unsupervised GRPO method, MM-UPT, cannot consistently improve performance across all benchmarks. As shown in Table 1, the performance on MathVerse drops modestly in some data settings. Can the authors explain the possible reasons for this inconsistency?
>
> **A1**: Thanks for your question. **We believe that this inconsistency is a natural and common phenomenon in self-improving settings** [1,2]. For instance, [1] reports that their self-improvement strategy improves LLaMA2’s performance on MMLU but leads to performance drops on ARC and HellaSwag. Similarly, [2] shows that their method enhances Mistral-7B on ARC and TruthfulQA, while slightly degrading its performance on MMLU. This reflects a broad trend in the field: self-training can be hard to guarantee the consistent performance gain across all benchmarks.
>
> In addition, as our unsupervised MM-UPT relies on a pseudo-reward from majority voting, the reliability of this reward signal is dependent on the model's initial accuracy and consistency on the training data. That said, **this reward signal may be noisy or suboptimal, particularly when the base model’s initial performance is weak on some MathVerse-style questions contained in the datasets like Geometry3k**. In such cases, MM-UPT may reinforce incorrect but confident outputs, leading to the modest performance drop in MathVerse when training on some datasets.
>
> [1] Yuan, Weizhe, et al. "Self-rewarding language models." ICML 2024.
>
> [2] Yang, Haoyan, et al. "Dynamic Noise Preference Optimization for LLM Self-Improvement via Synthetic Data."
>
> ---
>
> **Q2**. Based on Table 1, it seems that using different kinds of data brings significantly different improvements on different benchmarks. However, the paper lacks analysis on why different training data improve performance differently and lacks experiments investigating whether combining these datasets could bring additional performance improvements.
>
> **A2**: Thanks for your comment. We agree that different training datasets lead to different improvements across benchmarks, and we provide further explanation and supporting experiments below.
>
> First of all, this variation largely stems from the domain and structure alignment between the training data and evaluation tasks. For example, MMR1 provides more diverse multi-modal data formats (including diagrams, tables, and charts), while Geometry3K and GeoQA are more specialized towards geometric data. In addition, the pseudo-reward signal quality also plays a critical role in different datasets as mentioned in Question 1, which can bring significantly different improvements on different benchmarks.
>
> Second, **training with different datasets separately (rather than combining them) is a standard practice in finetuning**. For example, it is common to train models separately on GSM8K and MATH, as shown in [1]. This strategy allows for controlled analysis of transfer behavior and prevents data imbalance.
>
> Moreover, **we also conduct experiments where we combine Geometry3K, GeoQA, and MMR1 together for training, and observe that the model achieves further improvements in overall performance**. These results suggest that data combination is a viable strategy, especially when the datasets have complementary coverage.
>
> | Different Datsets | We-Math | MathVista | MathVerse | MathVision | **Avg**    |
> | ----------------- | ------- | --------- | --------- | ---------- | ---------- |
> | Geometry3k        | 0.6661  | 0.6850    | 0.4246    | 0.2733     | 0.5123     |
> | GeoQA             | 0.6822  | 0.6890    | 0.4368    | 0.2707     | 0.5197     |
> | MMR1              | 0.6874  | 0.7290    | 0.4487    | 0.2615     | 0.5316 |
> | Combine                  |  0.6970       | 0.7190          |   0.4492        |     0.2757       |  **0.5352**  |
>
>
>  [1] Lin, Zhihang, et al. "Cppo: Accelerating the training of group relative policy optimization-based reasoning models."
>
> ---
>
> **Q3**. Based on the basic idea of majority voting and the analysis of semantic entropy during training, this method encourages the model to converge toward more consistent predictions. My concern is that this may reduce response diversity. As a result, I'm wondering whether this method affects the model's performance on other non-math benchmarks?
>
> **A3**: Thank you for the question. We agree that our method tends to reduce output diversity by encouraging majority-agreement responses. However, this is not unique to MM-UPT. In fact, **response diversity reduction is a widely observed phenomenon in supervised reinforcement learning (RL) as well**: Some works [1,2] have shown that the reasoning capability boundary of LLMs often narrows as RLVR training progresses. Therefore, it is expected that MM-UPT, which operates under a similar reinforcement learning setup, would exhibit similar behavior.
>
> In our study, we mainly focus on multi-modal reasoning because it is a very challenging task. To investigate potential side effects on broader generalization, **we also extend our evaluation for the model trained on MMR1 using MM-UPT on two non-math benchmarks: ChartQA [3] and ICONQA [4]**, which test visual perception abilities over charts and natural images respectively. As shown below, MM-UPT does not hurt, and can even enhance performance on tasks outside math, likely due to the reinforcement of more reliable and consistent answer generation in general. This also suggests that the reduced diversity may not degrade model utility on other non-math benchmarks.
>
>
> | Models                    | ChartQA    | ICONQA     |
> | ------------------------- | ---------- | ---------- |
> | Qwen2.5-VL-7B-MM-UPT | **0.7748** | **0.5655** |
> | Qwen2.5-VL-7B             | 0.7196     | 0.5420     |
>
>
> [1] Yue, Yang, et al. "Does reinforcement learning really incentivize reasoning capacity in llms beyond the base model?."
>
> [2] Dang, Xingyu, et al. "Assessing diversity collapse in reasoning." SSI-FM@ICLR 2025.
>
> [3] Masry, Ahmed, et al. "Chartqa: A benchmark for question answering about charts with visual and logical reasoning."
>
> [4] Lu, Pan, et al. "Iconqa: A new benchmark for abstract diagram understanding and visual language reasoning."
>
> ---
>
> **Q4**. The method's effectiveness critically depends on the base model's initial accuracy being above chance (>0.5). This is a strong assumption that limits applicability, as the method cannot bootstrap knowledge on genuinely difficult or out-of-domain datasets where the model would initially have low performance.
>
> **A4**: Thank you for pointing out this. We agree that MM-UPT relies on the assumption that the base model achieves above-chance performance, which is necessary for majority voting to serve as a reliable pseudo-reward signal. **We have explicitly acknowledged and analyzed this limitation in Section 5.3. This is also a fundamental limitation shared by most unsupervised or self-improving frameworks**. Without a reasonably competent starting point, self-generated signals are likely to reinforce noise rather than useful knowledge. In this work, **we position MM-UPT as an effective post-training enhancement technique rather than a substitute for pretraining or supervised fine-tuning**. Nonetheless, we view this as a promising direction for future work. One avenue is to incorporate more robust reward estimation strategies (e.g., LLM-as-a-Judge, collaborative verification) that go beyond majority voting, allowing self-improvement even in low-confidence regimes
>
> ---
>
> **Q5**. Given that evaluation focuses solely on multi-modal mathematical reasoning, can this method transfer to improve pure language math tasks or other non-math tasks?
>
> **A5**: Thank you for the question. As discussed in A3, we have already evaluated MM-UPT on non-math multi-modal benchmarks (ChartQA and ICONQA), and observed consistent performance gains, suggesting that MM-UPT does not harm generalization beyond multimodal math tasks.
>
> To further assess transferability to pure language math tasks, we apply MM-UPT to a language-only mathematical model, Qwen2.5-MATH-7B [1], trained on the DAPO-17K dataset [2] using the same self-rewarding mechanism. We evaluate the resulting model on two popular benchmarks: MATH [3] and Omni-MATH [4]. As shown below, **MM-UPT leads to substantial improvements on both benchmarks for Qwen2.5-MATH-7B**. These results suggest that MM-UPT is not limited to multi-modal settings, and it can also be effectively extended to purely language-based reasoning tasks, provided the base model demonstrates sufficient initial competency.
>
> | Models                    | MATH     | Omni-MATH |
> | ------------------------- | -------- | --------- |
> | Qwen2.5-MATH-7B + MM-UPT    | **0.7580** | **0.3272** |
> | Qwen2.5-MATH-7B           | 0.5120   | 0.1809    |
>
>
> [1] Yang, An, et al. "Qwen2. 5-math technical report: Toward mathematical expert model via self-improvement."
>
> [2] Yu, Qiying, et al. "Dapo: An open-source llm reinforcement learning system at scale."
>
> [3] Hendrycks, Dan, et al. "Measuring mathematical problem solving with the math dataset."
>
> [4] Gao, Bofei, et al. "Omni-math: A universal olympiad level mathematic benchmark for large language models."

---

> > ### Comment · Area_Chair_nj6y · 2025-08-04
> > **Please respond to the author's rebuttal post**
> >
> > Hi Reviewer QjgG, I see no response letting me know whether or not the rebuttal has changed your opinion. Could you please let me and the authors know by engaging? This process is critical to enabling the (S)ACs to make a decision on this work.
> >
> > --Your AC

---

> > ### Comment · Reviewer_QjgG · 2025-08-09
> >
> > I appreciate for the author's response, which has addressed my concerns. I will keep my initial score.

---

> > > ### Author Response · Authors · 2025-08-09
> > >
> > > Thank you for your follow-up and for acknowledging our response. We're glad we were able to address your concerns. We sincerely appreciate the time and effort you've dedicated to this. Thanks again for your review and comments.

---

### Decision · Program_Chairs · 2025-09-17

**Decision:**

Accept (poster)

**Comment:**

The paper presents a simple method of performing a form of self play, wherein a policy rewards itself if multiple sampled trajectories agree via a majority vote.

- While the reviewers agree that the idea shows improvements in the order of ~3-5 absolute percentage points across a few visual math-based benchmarks, the empirical evidence does not appear to match the claims made.
- E.g. "a novel framework for continuous self-improvement without external supervision" is not backed up and is a *very strong* claim as even within their benchmarks, this method results in drops in some settings (this is also noted by reviewer QjgG) but the author's response that "this inconsistency is a natural and common phenomenon in self-improving settings" is directly at odds with the claims initially made.

If the claims are toned down to what the results actually show, then I would be ok with recommending acceptance as the execution of the idea has merits that other works could build off of, but in its current form - the paper promises more than it delivers.